# The architecture of EMC reveals a path for membrane protein insertion

John P O'Donnell[1†], Ben P Phillips[1†], Yuichi Yagita[1†], Szymon Juszkiewicz[1], Armin Wagner[2], Duccio Malinverni[1], Robert J Keenan[3], Elizabeth A Miller[1], Ramanujan S Hegde[1*]

[1]MRC Laboratory of Molecular Biology, Cambridge, United Kingdom; [2]Diamond Light Source, Didcot, United Kingdom; [3]Department of Biochemistry and Molecular Biology, The University of Chicago, Chicago, United States

**Abstract** Approximately 25% of eukaryotic genes code for integral membrane proteins that are assembled at the endoplasmic reticulum. An abundant and widely conserved multi-protein complex termed EMC has been implicated in membrane protein biogenesis, but its mechanism of action is poorly understood. Here, we define the composition and architecture of human EMC using biochemical assays, crystallography of individual subunits, site-specific photocrosslinking, and cryo-EM reconstruction. Our results suggest that EMC's cytosolic domain contains a large, moderately hydrophobic vestibule that can bind a substrate's transmembrane domain (TMD). The cytosolic vestibule leads into a lumenally-sealed, lipid-exposed intramembrane groove large enough to accommodate a single substrate TMD. A gap between the cytosolic vestibule and intramembrane groove provides a potential path for substrate egress from EMC. These findings suggest how EMC facilitates energy-independent membrane insertion of TMDs, explain why only short lumenal domains are translocated by EMC, and constrain models of EMC's proposed chaperone function.

*For correspondence:
rhegde@mrc-lmb.cam.ac.uk

†These authors contributed equally to this work

## Introduction

The endoplasmic reticulum (ER) is the site for biogenesis of nearly all eukaryotic integral membrane proteins (*Shao and Hegde, 2011a*). The defining feature of these proteins is the presence of one or more α-helical TMDs (*von Heijne, 2007*). Successful biogenesis requires each of these TMDs to be moved from the aqueous phase of the cytosol into the hydrophobic core of the lipid bilayer (*Guna and Hegde, 2018*; *White and von Heijne, 2005*). Although this insertion reaction can occur unassisted in vitro for some substrates (*Brambillasca et al., 2005*; *Brambillasca et al., 2006*), insertion in the crowded cellular environment typically requires factors that facilitate the reaction to minimize off-pathway outcomes such as aggregation, mislocalization, and degradation (*Anghel et al., 2017*; *Guna et al., 2018*; *Heinrich et al., 2000*; *Samuelson et al., 2000*; *Wang et al., 2014*).

The best understood insertion factor is the protein translocation channel formed by the heterotrimeric Sec61 complex (*Rapoport et al., 2017*). Structural studies have demonstrated that the Sec61α subunit contains an hourglass pore across the membrane for polypeptide translocation (*Van den Berg et al., 2004*; *Voorhees et al., 2014*). The wall of this pore contains a lateral gate that opens to provide hydrophobic domains in a substrate access to the lipid bilayer (*Gogala et al., 2014*; *Li et al., 2016*; *Voorhees and Hegde, 2016*). Thus, Sec61 is thought to facilitate TMD insertion by its distinctive architecture that connects the aqueous environment in the cytosol to the hydrophobic environment inside the membrane.

In addition to the Sec61 complex, the ER contains two other widely conserved insertases that both mediate the insertion of tail-anchored (TA) membrane proteins (*Guna and Hegde, 2018*). TA proteins contain a single TMD close to the C-terminus with a short unstructured domain translocated across the membrane (*Kutay et al., 1993*). This topology necessitates that the TMD is inserted post-

translationally. The 'guided entry of TA proteins' (GET) pathway (*Chio et al., 2017*; *Hegde and Keenan, 2011*) culminates at a heterodimeric complex (made of the ER-resident membrane proteins Get1 and Get2) that inserts TA proteins delivered to it by the targeting factor Get3 (*Mariappan et al., 2011*; *Wang et al., 2014*). More recently, the ten-subunit 'ER membrane protein complex' (EMC) (*Christianson et al., 2012*; *Jonikas et al., 2009*) was shown to insert TA proteins whose TMDs are insufficiently hydrophobic to effectively engage TRC40, the mammalian homolog of Get3 (*Guna et al., 2018*).

In addition to TA proteins, EMC mediates co-translational insertion of TMDs close to the N-terminus in the $N_{exo}$ topology (defined by a translocated N-terminus) (*Chitwood et al., 2018*). Notably, the translocated domain is short and unstructured. When the N-terminus is extended and preceded by a signal peptide, insertion is no longer EMC-dependent and occurs instead via the Sec61α lateral gate (*Chitwood et al., 2018*). Although the topology of $N_{exo}$ TMDs is opposite to the TMDs of TA proteins, they are both terminal TMDs whose insertion is not accompanied by appreciable polypeptide translocation. The EMC-mediated insertion reactions of both types of terminal TMDs has been reconstituted with purified EMC in vitro (*Chitwood et al., 2018*; *Guna et al., 2018*), suggesting that they might use similar mechanisms (*Chitwood and Hegde, 2019*).

The ten subunits of mammalian EMC (termed EMC1 through EMC10) are poorly understood because they have very few clearly established domains or resemblance to proteins of known structure or biochemical activity (*Wideman, 2015*; *Figure 1A*). The one possible exception is the three-TMD protein EMC3, which is predicted to be topologically and evolutionarily related to Get1 and a subdomain of the prokaryotic insertase YidC (*Anghel et al., 2017*). It has been speculated that this potential structural similarity reflects a similarity in molecular function.

The membrane subunit EMC3 is in complex with six other integral membrane EMC subunits (1, 4, 5, 6, 7, and 10) that together contain 12 predicted TMDs (*Chitwood and Hegde, 2019*; *Christianson et al., 2012*; *Wideman, 2015*). The seven membrane subunits of EMC associate with the cytosolic subunits EMC2, EMC8, and EMC9. EMC8 and EMC9 are ~44% identical in mammals, and not all species contain both genes (*Wideman, 2015*). Whether they are both part of a single 10-protein complex or substitute for each other in a 9-protein complex is not known. No free population of individual subunits has been detected (*Chitwood et al., 2018*; *Guna et al., 2018*; *Volkmar et al., 2019*), and disruption of most EMC subunits causes loss of EMC integrity and function (*Volkmar et al., 2019*). Thus, EMC is thought to function as a stable complex to mediate TMD insertion. Notably, this reaction appears to be energy independent in reconstitution assays in vitro (*Guna et al., 2018*), consistent with the absence of any nucleotide-binding domain in any of its subunits (*Wideman, 2015*).

EMC has also been suggested to act as a co-translational chaperone that captures individual or bundles of TMDs as they exit laterally from the Sec61 complex (*Shurtleff et al., 2018*). This postulated function has been inferred from analysis of EMC-mediated co-translational proximity biotinylation of ribosomes translating membrane proteins. How EMC might act as a chaperone, and the relationship of this function to its insertase activity, is not known. Consistent with either function, many membrane proteins are partially or strongly impacted in their biogenesis by the loss of EMC in numerous organisms including yeast, worms, flies, and mammals (*Bircham et al., 2011*; *Chitwood et al., 2018*; *Guna et al., 2018*; *Lakshminarayan et al., 2020*; *Louie et al., 2012*; *Richard et al., 2013*; *Satoh et al., 2015*; *Shurtleff et al., 2018*; *Talbot et al., 2019*; *Taylor et al., 2005*; *Volkmar et al., 2019*). Thus, EMC is a highly abundant component of the ER needed for cellular and organism homeostasis.

Understanding the function(s) of EMC requires knowledge of its structure. By analogy to the Sec61 complex, EMC's insertase function might involve a path from the cytosol into the membrane. To investigate this idea, we used a combination of biochemical, biophysical, and structural approaches to determine the architecture of EMC. Our findings suggest that the cytosolic subunits initially engage a TMD in a weakly hydrophobic vestibule that is contiguous with an intramembrane groove open to the lipid bilayer. This work provides a mechanistic framework for how the highly abundant and conserved EMC functions as a TMD insertase, explains why EMC acts preferentially on terminal TMDs, and suggests that EMC is unlikely to chaperone TMDs released at the Sec61α lateral gate.

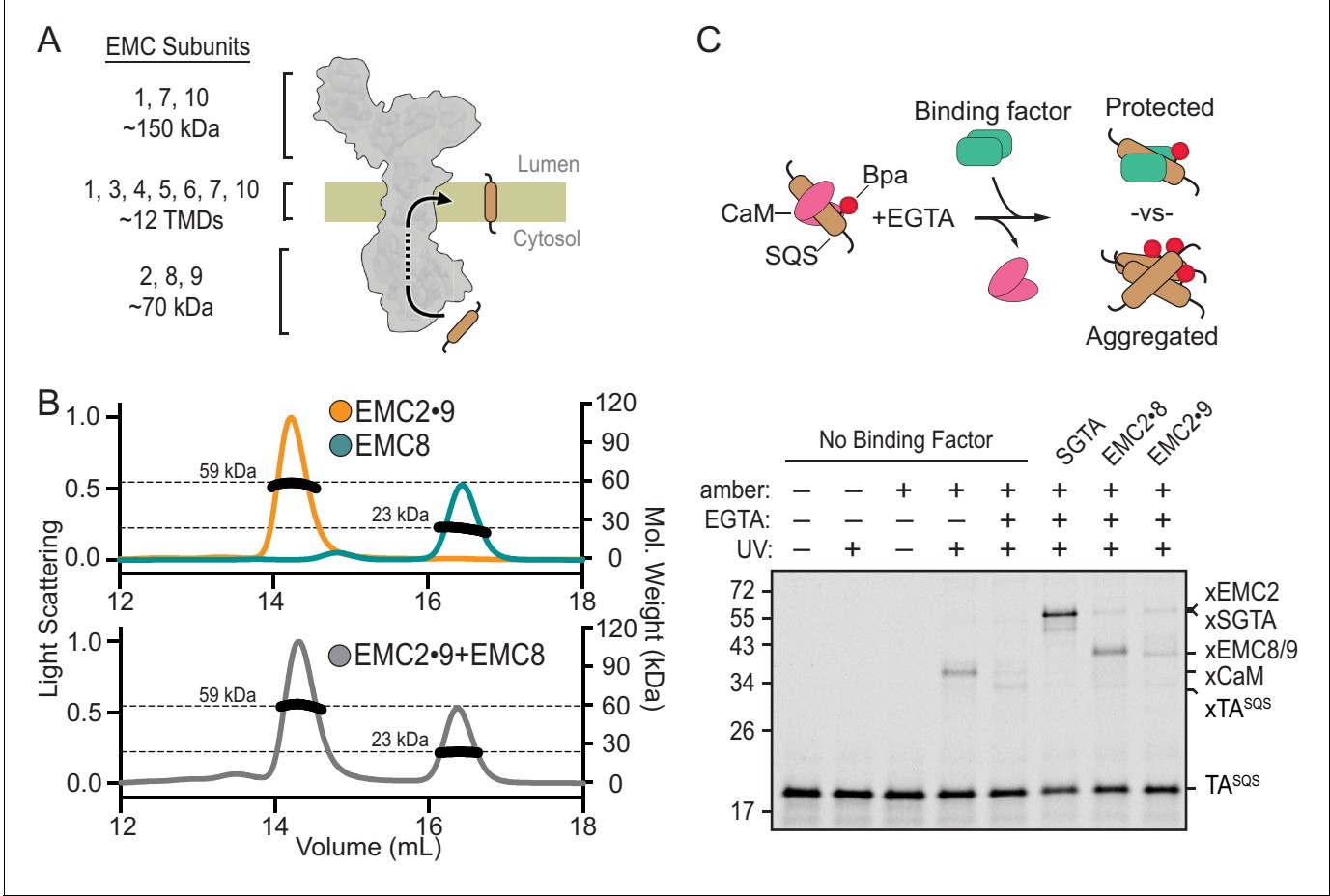

**Figure 1.** EMC2•EMC8 and EMC2•EMC9 form complexes that can bind a TMD. (**A**) Diagram of predicted EMC mass distribution and subunits. (**B**) Size exclusion chromatography coupled to multi-angle light scattering (SEC-MALS) analysis of recombinant EMC2•EMC9 reveals a stable complex in a 1:1 ratio at the expected molecular weight of 59 kDa (orange). Recombinantly expressed EMC8 analyzed independently has a Mw of 23 kDa consistent with a monomeric state (teal). Addition of EMC8 to a pre-formed EMC2•EMC9 complex does not result in the formation of a ternary complex (grey). (**C**) $^{35}$S-methionine-labeled TA$^{SQS}$ containing the benzoyl-phenylalanine (Bpa) photo-crosslinker within the TMD was produced as a defined complex with CaM using the PURE in vitro translation system. Bpa is incorporated into the TMD by amber suppression. Addition of EGTA releases CaM. The released TA$^{SQS}$ will either aggregate or be protected from aggregation by a TMD-binding protein. The outcome can be monitored by UV-mediated crosslinking via Bpa to nearby proteins. The position of crosslinks between TA$^{SQS}$ and various partners are indicated. No UV-mediated crosslinks are seen when TA$^{SQS}$ does not contain an amber codon (lanes 1 and 2).

The online version of this article includes the following figure supplement(s) for figure 1:

**Figure supplement 1.** SEC-MALS of individual and complexed cytoplasmic EMC-subunits.

**Figure supplement 2.** Activity of EMC cytosolic subunits in preventing TMD aggregation.

**Figure supplement 3.** EMC8 and EMC9 necessary for membrane protein biogenesis but are functionally redundant.

## Results

### Cytosolic EMC subunits can engage a substrate TMD

Purified native EMC reconstituted in proteoliposomes is sufficient to catalyze the insertion of a TA protein substrate containing the TMD from squalene synthase (SQS) (*Guna et al., 2018*). Because insertion of the SQS TMD into protein-free liposomes is comparatively inefficient, EMC is likely to transiently engage its substrate analogous to how other insertases interact with TMD substrates during insertion (*Klenner et al., 2008*; *Wang et al., 2014*; *Yu et al., 2008*). Reasoning that the first point of engagement by EMC might involve EMC's cytosolic subunits, we began by defining the interactions between EMC2, EMC8, and EMC9, then analyzing these proteins for their capacity to engage substrate.

Purified recombinant EMC2 formed a stable complex with either EMC8 or EMC9. Size exclusion chromatography coupled to multi-angle light scattering (SEC-MALS) showed that each individual protein is monomeric and the EMC2•EMC8 and EMC2•EMC9 complexes are heterodimers (*Figure 1B*; *Figure 1—figure supplement 1*). The EMC2•EMC9 heterodimer did not form a ternary complex with excess EMC8, and the EMC2•EMC8 heterodimer did not form a ternary complex with excess EMC9. Thus, the cytosolic domain of EMC is likely to be composed of EMC2 in complex with either EMC8 or EMC9, but not both. The presence of only one of either EMC8 or EMC9 in native EMC may explain why some species have only one of these two genes (*Wideman, 2015*).

To analyze substrate interaction, a $^{35}$S-labeled TA protein containing the TMD of SQS (TA$^{SQS}$) was produced in vitro using a fully purified translation system derived from *E. coli* components (*Shimizu and Ueda, 2010*). The photocrosslinking amino acid 4-Benzoylphenylalanine (Bpa) was incorporated within the TMD by amber suppression and TA$^{SQS}$ was kept soluble by including an excess of the chaperone-like protein calmodulin (CaM) (*Guna et al., 2018*; *Shao and Hegde, 2011b*). UV irradiation of this complex produced a TA$^{SQS}$-CaM crosslinked product which was diminished if CaM was inactivated by chelation of Ca$^{2+}$ with EGTA (*Figure 1C*, lanes 4, 5). Without a chaperone, SQS formed self-crosslinks due to its aggregation as documented previously (*Guna et al., 2018*). Release of SQS from CaM in the presence of EMC2•EMC8 or EMC2•EMC9 showed crosslinks to EMC2, EMC8, and EMC9 concomitant with a reduction of the TA$^{SQS}$ aggregate crosslink (*Figure 1C*, lanes 7,8). This effect of the EMC2•EMC8 and EMC2•EMC9 complexes is similar, but less complete, than the effect seen with the TMD chaperone SGTA (*Shao et al., 2017*; *Figure 1C*, lane 6). Analysis of individual subunits at various concentrations showed that EMC2 inhibited aggregation with a similar potency as the heterodimeric complexes, whereas EMC8 showed somewhat lower potency and EMC9 was largely inert (*Figure 1—figure supplement 2*).

These observations indicate that the cytosolic domain of EMC contains a TMD-binding heterodimer of the EMC2•EMC8 subcomplex or the homologous EMC2•EMC9 subcomplex. Interchangeability of EMC8 and EMC9 explains why knockdown of EMC8 is accompanied by increased EMC9 (*Volkmar et al., 2019*), and why individual knockdowns do not impact EMC substrates but the double-knockdown does (*Figure 1—figure supplement 3*). Whether there are functional differences between EMC8- versus EMC9-containing EMC for certain substrates remains to be determined.

## Crystal structure of the EMC2•EMC9 complex

Recombinant cytosolic subunits and subcomplexes were screened for crystal formation, resulting in well-diffracting crystals of a nearly full length complex containing EMC2 and EMC9. This complex, lacking only short regions at the termini, was verified to be functional for substrate interaction by the photo-crosslinking assay (data not shown). The EMC2•EMC9 structure was solved with experimental phases from a single-wavelength anomalous diffraction (SAD) experiment using endogenous sulphur atoms for anomalous signal (*Wagner et al., 2016*). An initial model was built de novo and subsequently used as a molecular replacement search model for a native crystal diffracting X-rays to 2.2 Å (*Table 1*).

EMC2 is largely alpha-helical and contains a curved tetratricopeptide repeat (TPR) motif buttressed on one side by EMC9 (*Figure 2A*). The core of EMC9 consists of a small β-barrel flanked by alpha-helices. The ~1100 Å$^2$ EMC2-EMC9 interface contains a network of hydrogen bonds and salt bridges (*Krissinel and Henrick, 2007*), explaining its high stability in vitro and the absence of any appreciable free population of either protein in cells. EMC2 and EMC9 both contribute to the formation of a large, relatively shallow and moderately hydrophobic cavity that we term the cytosolic vestibule (*Figure 2B*).

A reference-free model of EMC8 predicted by Rosetta (*Rohl et al., 2004*) possessed the same core fold as the EMC9 structure with a RMSD of 2.3 Å. When EMC9 was used as a reference (*Song et al., 2013*), the alpha-carbon backbone of the EMC8 model was indistinguishable from EMC9 with an RMSD of 0.6 Å (*Figure 2—figure supplement 1*). This observation is consistent with their high homology (~44% identity) and comparable affinity for EMC2 (*Figure 2—figure supplement 2*). We therefore conclude that the EMC2•EMC8 structure is likely to be very similar to EMC2•EMC9 explaining why the absence of either one has no obvious phenotype in cells.

**Table 1.** X-ray data collection and refinement statistics.

| | EMC2•EMC9 | EMC2•EMC9 |
|---|---|---|
| | (S-SAD phasing) | (Molecular replacement) |
| **Data collection** | | |
| X-ray source | Diamond I23 | Diamond I03 |
| X-ray wavelength (Å) | 2.7552 | 0.9763 |
| Space group | $P2_12_12_1$ | $P2_12_12_1$ |
| **Unit cell parameters** | | |
| a, b, c (Å) | 53.2 82.8 124.0 | 52.6 84.9 122.6 |
| α, β, γ (°) | 90.0 90.0 90.0 | 90.0 90.0 90.0 |
| Resolution range (Å) | 49.6–2.65 (2.78–2.65) | 49.7–2.20 (2.27–2.2) |
| **No. of reflections** | | |
| Total | 668148 (73113) | 1486886 (128644) |
| Unique | 16468 (2109) | 28668 (2433) |
| Completeness (%) | 99.7 (98.9) | 100.0 (100.0) |
| Multiplicity | 40.6 (34.7) | 51.9 (52.9) |
| I/σ(I) | 36.8 (1.8) | 17.6 (1.9) |
| $R_{meas}$ (%) | 6.4 (214.6) | 17.2 (465.7) |
| $R_{merge}$ (%) | 6.3 (211.5) | 17.0 (461.3) |
| $R_{pim}$ (%) | 1.0 (35.3) | 2.4 (63.7) |
| $CC_{1/2}$ (%) | 100.0 (81.7) | 100.0 (92.6) |
| **Refinement** | | |
| $R_{work}/R_{free}$ (%) | - | 20.3/25.0 |
| **RMS deviations** | | |
| Bond length (Å) | - | 0.007 |
| Bond angle (°) | - | 0.848 |
| **No. of atoms** | | |
| Protein | - | 3454 |
| Ligands | - | 72 |
| Water | - | 80 |
| **Average B-factors (Å$^2$)** | | |
| Total | - | 70.6 |
| Protein | - | 71.7 |
| Ligands and waters | - | 65.4 |
| **Ramachandran (%)** | | |
| Favored | - | 96.4 |
| Outliers | - | 0.0 |
| PDB Code: | - | 6Y4L |

*(*) Values in brackets are for the highest resolution bin.

## Functional analysis of the vestibule in the EMC2•EMC9 complex

Based on the ability of a substrate TMD to photo-crosslink with both EMC2 and EMC9, we posited that a TMD might bind to the vestibule in the EMC2•EMC9 complex. The moderate hydrophobicity of this region in both the EMC2•EMC9 structure and EMC2•EMC8 model is consistent with the moderate potency of these complexes in preventing substrate aggregation in aqueous solution. To test whether substrate binds in this cavity, we prepared the EMC2•EMC9 complex containing the photo-crosslinker Bpa at various sites and tested UV-mediated crosslinking to the TMD of SQS (*Figure 3A and B*; *Figure 3—figure supplement 1*). In this experiment, we used a substrate that contains only

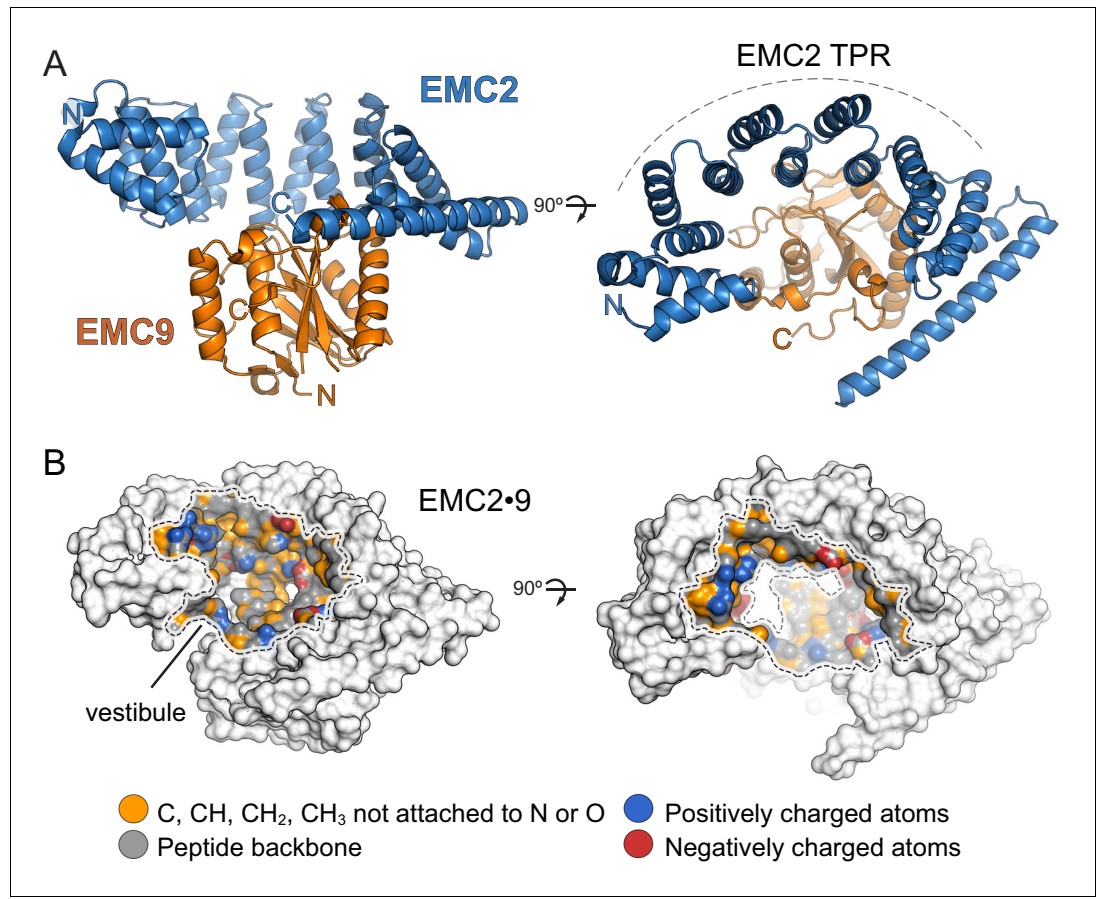

**Figure 2.** Structure of the EMC2•EMC9 complex. (**A**) Crystal structure of the EMC2•EMC9 heterodimer (PDB: 6Y4L). The heterodimer consists of EMC2 (residues 11–274), depicted in blue, and EMC9 (residues 1–200), depicted in orange. The TPR-repeat motif of EMC2 is indicated. (**B**) Physicochemical properties of a vestibule in the EMC2•EMC9 complex. Surface rendering of crystal structures coloured according to chemical properties (*Hagemans et al., 2015*).

The online version of this article includes the following figure supplement(s) for figure 2:

**Figure supplement 1.** Protein structure prediction confirms structural homology between EMC9 and EMC8.
**Figure supplement 2.** EMC8 and EMC9 have similar affinities for EMC2.

the TMD and a few flanking residues (termed TMD$^{SQS}$), allowing us to be sure that the crosslinks are to the TMD. These findings verify that the cavity can engage substrate TMDs and map an approximate binding site that spans both EMC2 and EMC9.

To test the functional relevance of the vestibule, we perturbed its physicochemical properties at regions that form relatively strong versus relatively weak crosslinks with substrate and analyzed EMC function in cells. Due to the redundancy of EMC2•EMC8 and EMC2•EMC9 heterodimers, we focused our mutational analysis on EMC2. Various residues in EMC2 were mutated individually or in combination to charged amino acids and tested for insertion of a SQS-based reporter in cultured cells (*Figure 3C*; *Figure 3—figure supplement 2A*). In this flow cytometry assay, failed insertion of RFP-tagged SQS results in a lower RFP signal due to its degradation relative to an internal GFP control (*Guna et al., 2018*). As reported previously for other EMC subunit knockouts (*Guna et al., 2018*; *Volkmar et al., 2019*), the RFP:GFP ratio was very low in ΔEMC2 U2OS cells, but restored to normal levels when wild type EMC2 was re-expressed (*Figure 3C*, left graph).

A combination of H189E, L190E, and Y191K (termed HLY*) in the region of highest substrate crosslinking showed reduced insertion by this assay (*Figure 3C*, left graph). Individual mutations of these residues had little or no effects (*Figure 3—figure supplement 2A*). Mutation of adjacent residues that were already hydrophilic (Q193E and Q194K) showed no effect (*Figure 3—figure*

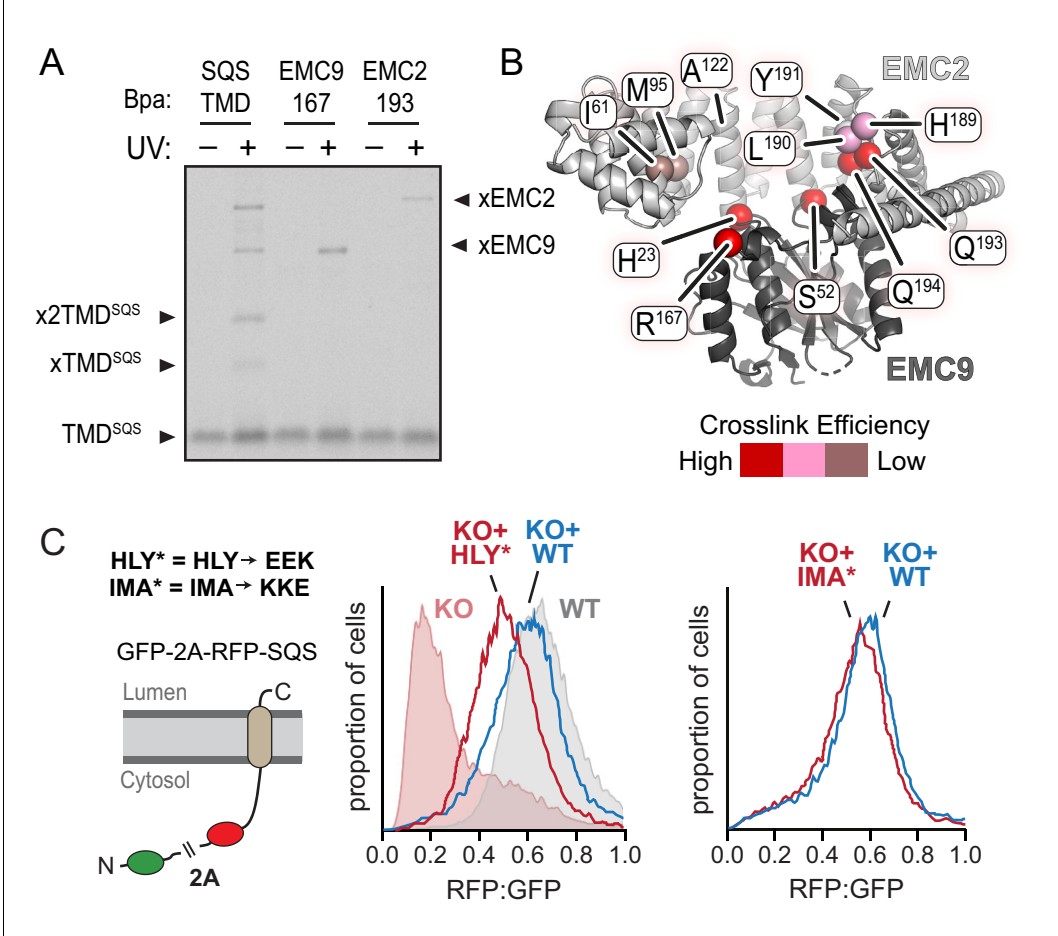

**Figure 3.** Functional analysis of the EMC2•EMC9 cytosolic vestibule. (**A**) [35]S-methionine-labeled TMD of SQS was mixed with recombinant purified EMC2•EMC9 complex as in *Figure 1* and analyzed directly or after UV irradiation as indicated. The photocrosslinking amino acid benzoyl-phenylalanine (Bpa) was incorporated into either the SQS substrate, EMC9 (at codon 167), or EMC2 (at codon 193) as indicated. Both R167 in EMC9 and Q193 in EMC2 line the vestibule. (**B**) Bpa was incorporated at different positions within the vestibule of the EMC2•EMC9 complex and crosslinking efficiency to SQS was determined as in panel A. Locations of the Bpa are annotated as spheres on the EMC2•EMC9 heterodimer. The sphere colors correspond to the intensity of the resulting crosslink. Position 191 (obscured behind L190 in this view) showed no crosslinking to substrate, consistent with its rearward facing location. (**C**) Shown on the left is a diagram of the dual color reporter for insertion of the TMD of SQS. Expression of this reporter results in a free GFP protein and an RFP-tagged SQS protein due to ribosomal skipping at the a viral 2A sequence. The left graph shows flow cytometry analysis of the SQS reporter in WT cells (grey), EMC2 knockout (KO) cells (shaded pink), KO cells complemented with WT EMC2 (blue line), and KO cells complemented with the HLY* EMC2 mutant (red line). The data are represented as histograms of the RFP to GFP ratio. The right graph shows a comparison of the SQS reporter in KO cells complemented with either WT EMC2 or the IMA* EMC2 mutant. The mutated amino acids, whose positions are shown in panel B, are: H189E, L190E, Y191K, I61K, M95K, and A122E.

The online version of this article includes the following figure supplement(s) for figure 3:

**Figure supplement 1.** Crosslinking analysis of the EMC2•EMC9 cytosolic vestibule.

**Figure supplement 2.** Functional analysis of the EMC2•EMC9 cytosolic vestibule.

---

*supplement 2A*), suggesting that a substantial change in hydrophobicity or possibly a change in local conformation was needed to appreciably impact substrate insertion. The IMA* mutation, which causes a similarly large loss of hydrophobicity to the region of EMC2's vestibule where substrate crosslinking is weak, showed almost no impairment in substrate insertion (*Figure 3C*, right graph). Importantly, we verified that the HLY* mutation did not impair EMC2 incorporation into the native

EMC (*Figure 3—figure supplement 2B*), unlike the A129K mutation whose strong phenotype (*Figure 3—figure supplement 2A*) could be ascribed to poor assembly (*Figure 3—figure supplement 2C*). Thus, there is concordance between EMC2 regions of the vestibule that interact with a TMD substrate in vitro and mutations that perturb substrate insertion in cells.

## Position of the EMC2•EMC9 subcomplex within native EMC

Once a TMD substrate binds to the cytosolic subunits of EMC, subsequent insertion requires access to the lipid bilayer. To understand how the substrate-binding cavity within the cytosolic subunits is oriented relative to the membrane, we sought to place our EMC2•EMC9 structure within the architecture of native EMC. Affinity-purified EMC representing a mixture of EMC8- and EMC9-containing complexes (*Guna et al., 2018*) were analyzed by single-particle cryo-EM. The map clearly shows density for the lumenal, transmembrane, and cytoplasmic regions of EMC (*Figure 4A*). Due to preferential orientation, the resulting density map was limited to modest resolution of (6.5 Å) throughout the structure (*Table 2*; *Figure 4—figure supplement 1*).

Although atomic models could not be built de novo from the EM map, this resolution was sufficient to dock the EMC2•EMC9 crystal structure. The only region of EMC2•EMC9 that did not precisely align with the EM-density was the first three alpha-helices of EMC2 comprising residues 11–66 (*Figure 4—figure supplement 2*). Low frequency normal mode analysis (*Suhre and Sanejouand, 2004*) predicted that these three helices undergo structural movement that would be compatible with the EM-density. Therefore, the EMC2•EMC9 structure was refined against the EM-density using Flex-EM, (*Topf et al., 2008*), *Coot* (*Emsley et al., 2010*), and PHENIX real-space refinement (*Afonine et al., 2018*), resulting in a slightly rotated position that fits into the EM-density (*Figure 4B*; *Figure 4—figure supplement 2*).

The plane of the membrane was evident from the detergent micelle surrounding the TMD region of EMC (*Figure 4A*). Relative to the membrane, the EMC2•EMC9 complex is oriented such that the TPR-repeats of EMC2 are proximal to the membrane but angled at ~30˚. In this configuration, the substrate binding cavity of EMC2•EMC9 has access to both the bulk cytosol and the membrane domain of EMC (*Figure 4B*). The surface of EMC2 that faces the membrane domain is also highly conserved, consistent with this region making contacts with the membrane-embedded subunits of EMC (*Figure 4C*). Thus, the cytosolic subunits of EMC are arranged so the cavity capable of binding substrate forms a vestibule that links the cytosol to the integral membrane subunits that would act next to mediate TMD insertion.

The region of the vestibule that binds substrates as determined in crosslinking assays is occupied in the cryo-EM map by density that is contributed from another EMC subunit (possibly EMC6, as discussed below). Intramolecular placeholders that temporarily shield the substrate-binding pockets are also observed in the membrane protein targeting factors SRP and Get3 (*Mateja et al., 2015*; *Voorhees and Hegde, 2015*). In both of these other examples, the placeholders are less hydrophobic than substrate TMDs, allowing their displacement by bona fide substrates but presumably not other proteins. EMC may therefore operate similarly. Thus the putative placeholder density might provide an approximation of what a substrate-bound intermediate of EMC looks like. From this position, an inserting substrate would next have to engage the region of EMC embedded in the membrane.

## Architecture of the membrane-embedded and lumenal regions of EMC

A cross section through the detergent micelle of the EMC map in the plane of the membrane showed the arrangement of thirteen putative TMD helices (*Figure 5A*). To assign the intramembrane densities to individual EMC subunits, we first generated starting models for those that contain two or more TMDs. Using trRosetta (*Yang et al., 2020*), which employs co-evolutionary data, deep learning, and inter-residue contacts for energy minimization of structural models, we produced models for EMC3, EMC4, EMC5, and EMC6 (*Figure 5—figure supplement 1*). Even though trRosetta does not consider biological membranes or topology, the predicted TMDs of EMC3 and EMC5 pack together as helices whose lengths match the thickness of a lipid bilayer. Strikingly, trRosetta accurately predicted the structures of EMC2, EMC9, and Sec61α, providing confidence in its capacity to produce starting models for both soluble and integral membrane domains.

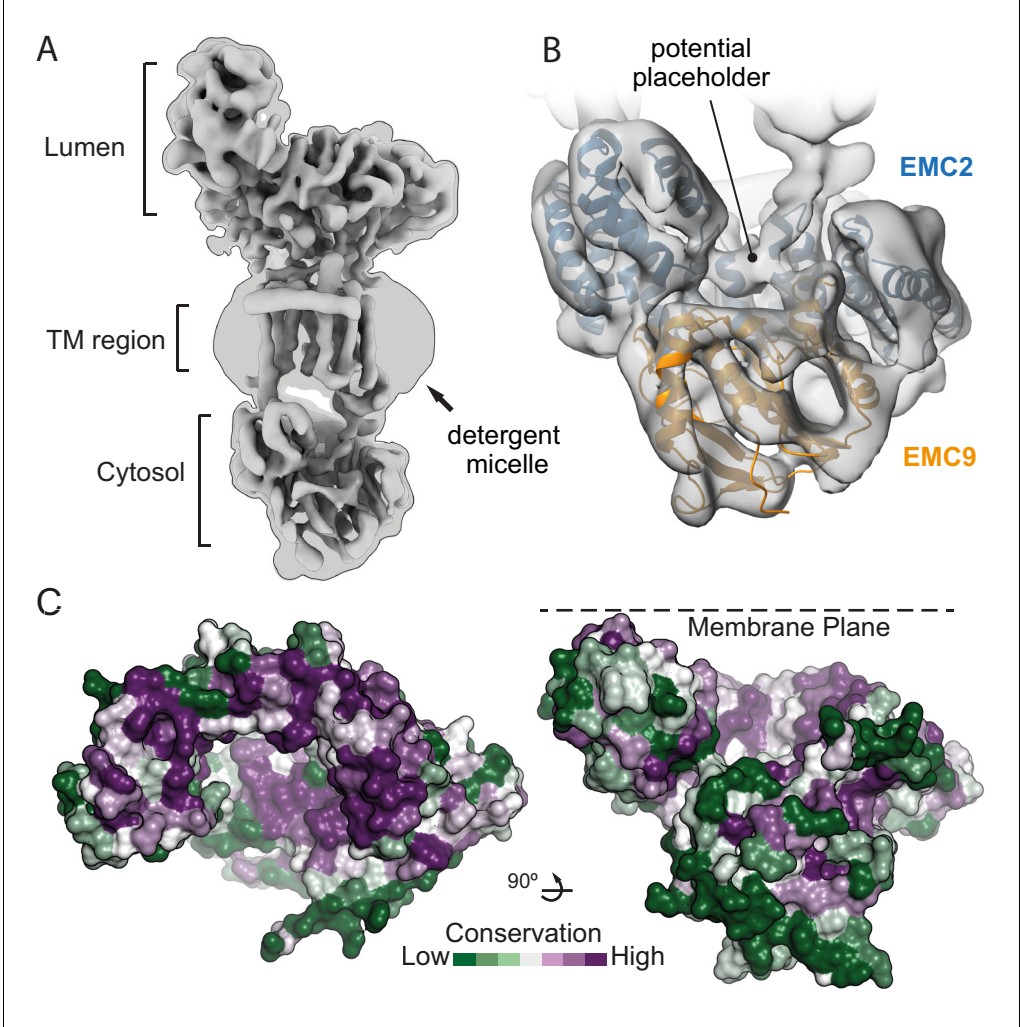

**Figure 4.** The position of EMC2•EMC9 within native EMC. (**A**) Cryo-EM map of the EMC at 6.4 Å resolution reveals the architecture of the complex. The map is shown at two contour levels: a stringent contour that illustrates secondary structure features (0.21) superimposed with a liberal contour that shows the detergent micelle (0.15). (**B**) Refinement of the EMC2•EMC9 crystal structure (blue and orange) into the cytosolic density (grey) using Flex-EM, Coot and PHENIX. (**C**) Surface rendering of the EMC2•EMC9 crystal structure coloured by residue conservation (*Ashkenazy et al., 2016*) from highly conserved (purple) to weakly conserved (green). The top rim of EMC2's TPR is highly conserved. This surface faces the membrane and regions of it interact with other EMC-subunits. The substrate binding vestibule also exhibits high conservation in comparison to the remaining solvent exposed surface of the EMC2•EMC9 heterodimer.

The online version of this article includes the following figure supplement(s) for figure 4:

**Figure supplement 1.** Cryo-electron microscopy data processing.

**Figure supplement 2.** Normal mode analysis and flexible fitting of the EMC2•EMC9 crystal structure into the full EMC cryo-EM map.

Although we had previously predicted two TMDs for EMC6 based on hydrophobicity profiles (*Krogh et al., 2001*), trRosetta generated a three-helix bundle (*Figure 5—figure supplement 1*) that matched the consensus of other topology prediction algorithms (*Tsirigos et al., 2015*). More surprisingly, EMC4 also is predicted by trRosetta to have three TMD-like helices, not two as previously thought based on topology algorithms. Protease-protection assays resolved this discrepancy in favor of the trRosetta model because we found that the N-terminus of EMC4 faces the cytosol and the C-terminus is in the ER lumen (*Figure 5—figure supplement 2*).

**Table 2.** Cryo-EM data collection and processing.

| | Dataset 1 | Dataset 2 | Dataset 3 | Dataset 4 | Dataset 5 |
|---|---|---|---|---|---|
| Microscope | Titan Krios (m06 eBIC) | Titan Krios (m06 eBIC) | Titan Krios (m06 eBIC) | Titan Krios (MRC-LMB) | Titan Krios (MRC-LMB) |
| Pixel Size | 1.380 | 1.380 | 1.380 | 1.179 | 1.390 |
| Voltage | 300 | 300 | 300 | 300 | 300 |
| Spherical Aberation | 2.7 | 2.7 | 2.7 | 2.7 | 2.7 |
| Total exposure (e$^-$/Å$^2$) | 39.60 | 42.50 | 37.77 | 39.36 | 44.36 |
| Exposure Length (s) | 5.0 | 11.02 | 14 | 11 | 11 |
| Frames | 25 | 44 | 40 | 44 | 40 |
| Defocus Range (μm) | −0.5 to −1.5 | −0.5 to −1.5 | −0.5 to −1.5 | −0.5 to −1.5 | −0.5 to −1.5 |
| Micrographs | 2776 | 2484 | 1206 | 4228 | 932 |
| Microscope tilt (degrees) | 0 | 0 | 30 | 20 | 20 |
| Volta Phase Plate | ✓ | ✓ | ✓ | ✓ | ✓ |
| **Pre-merge processing** | **Dataset 1** | **Dataset 2** | **Dataset 3** | **Dataset 4** | **Dataset 5** |
| Motion Correction and CTF estimation (micrographs) | 2776 | 2484 | 1206 | 4228 | 932 |
| Blob-based autopicking (particles) | 1,298,488 | 541,233 | 319,989 | 1,145,157 | 223,613 |
| 2x iterations of 2D classification (particles) | 63,180 | 63,315 | 51,371 | 305,685 | 28,662 |
| Template based autopicking (particles) | 467,323 | 687,645 | 826,006 | N/A | 273,852 |
| 2D classification (particles) | 11,862 | 71,977 | 63,881 | 113,852 | 66,799 |
| Ab Initio 3D classification (particles) | 103,105 | 71,977 | 46,349 | 113,852 | 50,232 |
| **Post-merge processing** | | | **Combined Datasets** | | |
| 2x Iterations ab initio 3D classification (particles) | | | 405,515 | | |
| Non-uniform Refinement (particles) | | | 167,294 | | |
| Per-particle CTF refinement | | | 6.71 Å map | | |
| Non-uniform refinement with local resolution estimation and filtering | | | 6.4 Å map | | |
| EMDB Deposition code | | | EMD-11058 | | |

The trRosetta models for EMC3, EMC6, and EMC5 could be docked at distinctive positions into the EM map based on their helix lengths and relative tilts (*Figure 5B*). The main remaining region that could accommodate a three-helix bundle was therefore assigned to EMC4. This left two TMD-like densities and three single-spanning EMC subunits (EMC1, EMC7, and EMC10). Because EMC10 can be depleted with no functional consequences (*Volkmar et al., 2019*), we suspected this subunit was probably peripheral and least likely to be visualized in the map. We therefore assigned the two remaining TMD-like densities to EMC1 and EMC7, distinguishing between them using a site-specific photo-crosslinking approach.

The photo-crosslinking amino acid 3'-azibutyl-N-carbamoyl-lysine (AbK) was introduced by amber suppression (*Ai et al., 2011*) into specific sites of individual FLAG-tagged EMC subunits. The cells were then UV-irradiated to induce crosslinks with nearby proteins (with a backbone-to-backbone distance of ~10–15 Å) and the samples were analyzed by immunoprecipitation and immunoblotting. Three and four sequential positions were tested within a TMD to sample different radial directions (*McCormick et al., 2003*). Introduction of AbK at position I23[AbK] in the first TMD of EMC3 showed a strong high-molecular weight crosslink consistent with the size of EMC1. EMC1 immunoblotting of natively purified EMC via the FLAG-tagged EMC3 verified that the UV-dependent EMC3 crosslinking partner is EMC1. Denaturing IP of EMC1 followed by anti-FLAG immunoblotting further validated this assignment. The EMC1 crosslink is sharply diminished at positions T24[AbK] and F25[AbK], but is partially recovered at position F26[AbK], thereby defining the face of this TMD helix that is adjacent to EMC1 (*Figure 5C*).

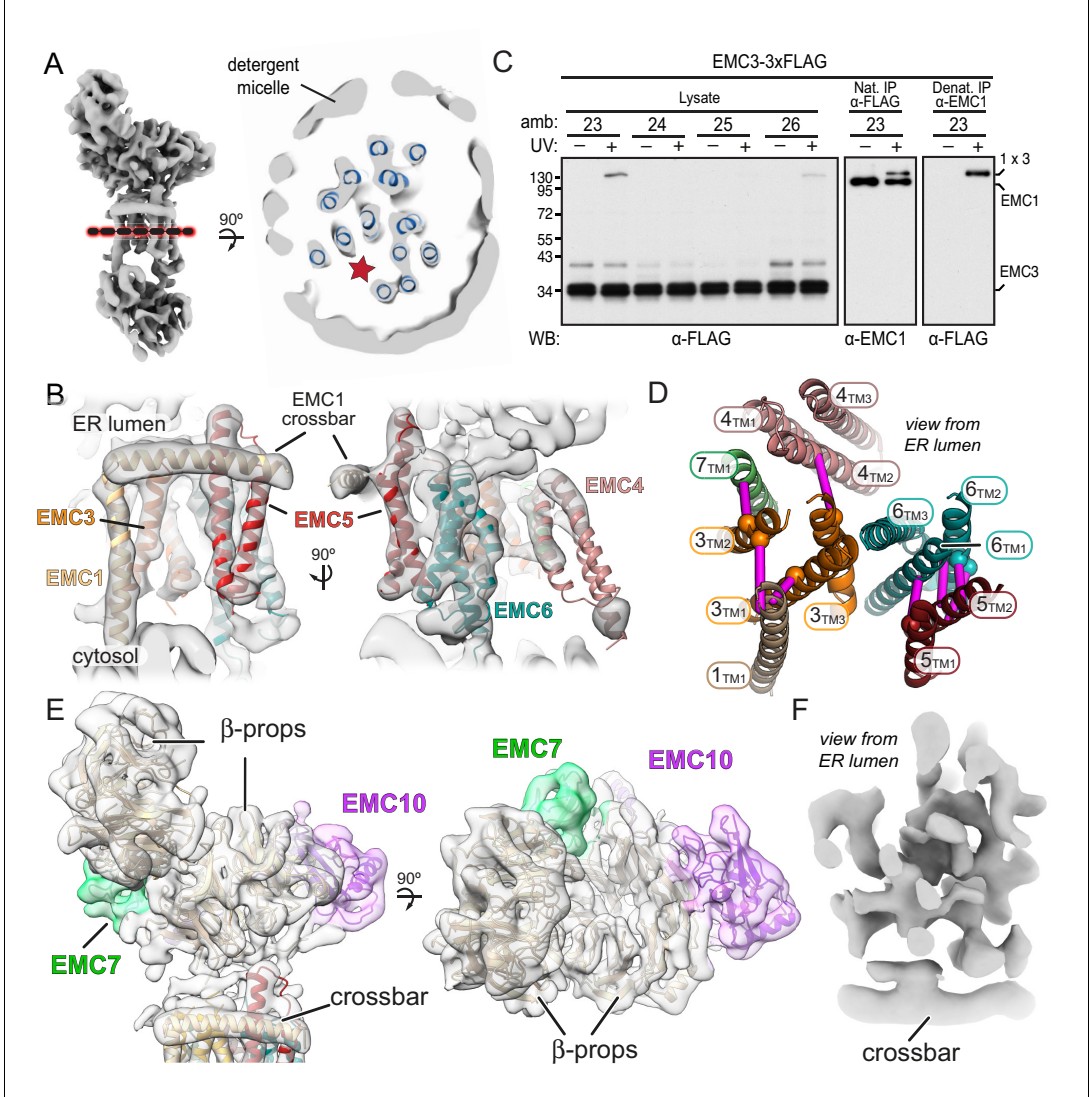

**Figure 5.** A composite model of EMC's membrane and lumenal domains. (**A**) Cross section through the TMD region of the cryo-EM map of EMC (light grey) identifies 13 helix-like densities (dark blue) that define an intramembrane groove (red star) open to the lipid bilayer. Density corresponding to the annular detergent micelle is indicated. (**B**) Ab initio trRosetta models fitted into cryo-EM density for the EMC membrane domain are shown for EMC1 (wheat), EMC3 (orange), EMC4 (light pink), EMC5 (red), EMC6 (teal), and EMC7 (green). The horizontal density at the membrane-lumen interface termed the 'crossbar' is assigned to an amphipathic helix in EMC1. (**C**) EMC3-3xFLAG constructs containing amber codons (amb.) at the indicated positions were expressed together with an amber suppressor tRNA and cognate aminoacyl-tRNA synthetase that accepts the UV-activated crosslinking amino acid 3'-azibutyl-N-carbamoyl-lysine (AbK). Cells were left untreated or irradiated with UV and analyzed by immunoblotting for EMC3-3xFLAG. A prominent UV-dependent crosslink is seen from position 23 and to a lesser extent, position 26. Native FLAG immunoprecipitation (IP) recovers EMC1 (indicating that EMC3-3xFLAG is incorporated into EMC), which shifts with UV. Denaturing EMC1 IP confirms the crosslinked product contains both EMC1 and EMC3. (**D**) TMD helices positioned based on docking of ab initio models overlayed with AbK-mediated crosslinks (see *Figure 5—figure supplement 3*). The positions where AbK was incorporated are shown as spheres, with magenta lines showing the closest point of the target protein in the model. (**E**) Composite model of the EMC lumenal domain generated by ab initio modelling in trRosetta and real-space refinement in PHENIX. Cryo-EM density has been colored according to subunit identity and the composite lumenal domain model accounts for almost all the lumenal EM density. (**F**) Cross-section through EMC at the plane of the membrane-lumen interface illustrating that a pore is not evident across the membrane. All EM data visualized in UCSF ChimeraX with EM maps contoured at 0.15 (panel A) and 0.21 (panels B, E, and F) with hide dust setting of 10. Panel D was generated in PyMOL.

The online version of this article includes the following figure supplement(s) for figure 5:

**Figure supplement 1.** Ab initio prediction of EMC subunit structure and flexible fitting using real-space refinement.

**Figure supplement 2.** Protease-protection analysis of EMC4 topology.

**Figure supplement 3.** Site-specific photocrosslinking between EMC subunits.

**Figure supplement 4.** Provisional assignment of non-EMC2•EMC9 cytosolic density.

*Figure 5 continued on next page*

*Figure 5 continued*

**Figure supplement 5.** Views of the composite EMC model.

Similar experiments placing AbK in other parts of EMC3 showed that TMD2, but not TMD3, is near EMC1 (*Figure 5—figure supplement 3*; results summarized in *Figure 5D*). These results not only validated the trRosetta model for EMC3, but also assigned the position of EMC1's sole TMD within the EM density. The remaining TMD-like density in the EM map was assigned to the TMD of EMC7 and supported by Abk-mediated crosslinks seen from TMD2 of EMC3. Additional crosslinking data between EMC5 and EMC6 supported our overall placements for the TMD regions of EMC subunits, resulting in a provisional model for the membrane domain of EMC (*Figure 5D*).

The EM density in the cytosolic domain contains regions not accounted by EMC2 or EMC8/9 (*Figure 5—figure supplement 4*). The regions of additional density in the cytosolic vestibule extend from the TMDs of EMC1 and possibly EMC6. Consistent with these being core features of EMC, the C-terminal tail of EMC1 and the N-terminal tail of EMC6 are highly conserved. Most of the remaining density could be fitted with minor adjustments to the trRosetta model for the cytosolic domains of EMC3. In particular, a three-helix bundle formed of the coiled-coil between TMD1 and TMD2 in complex with a helix in the C-terminal tail fit well into a pyramidal density on the backside of EMC2. This three-helix bundle is predicted based on co-evolution analysis using trRosetta. Consistent with this placement, AbK positioned in the EMC3 coiled coil where it approaches the three-helix bundle of EMC4 forms a UV-mediated EMC3-EMC4 crosslink. The remainder of the cytosolic density might correspond to EMC3's C-terminal tail. Thus, EMC2 organizes EMC's membrane subunits via its multi-pronged interactions with EMC1, EMC3, and EMC6, explaining why the membrane domain of EMC falls apart completely in the absence of EMC2 (*Volkmar et al., 2019*).

The lumenal region of EMC is almost entirely composed of the lumenal domains of EMC1, EMC7, and EMC10. Predictions of their structures (*Figure 5—figure supplement 1*) generated models that could be unambiguously docked into the lumenal region of the EM map (*Figure 5E*). In these models, EMC1 contains two beta-propellers, while EMC7 and EMC10 each contain a small beta-barrel of different sizes. Using these distinctive features to guide docking, other regions of continuous density could be assigned to other parts of the predicted models with minor adjustments. The most prominent additional feature was a long conserved amphipathic helix flanked by two long unstructured linkers. We assigned this EMC1 helix to a conspicuous density (termed the crossbar) that sits at the interface between the lumen and membrane. Inspection of the density at the plane of the crossbar where the TMDs interact with the lumenal domain shows that EMC does not have a pore across the membrane that connects the cytosolic vestibule to the ER lumen (*Figure 5F*).

In sum, all of the membrane-embedded and lumenal density in the EM map could be accounted for by rigid body fitting the independently generated trRosetta models (*Figure 5—figure supplement 1*). These models were subsequently real-spaced refined into the density using PHENIX. The combination of atomic resolution structures, contact-informed modeling, photocrosslinking, and moderate resolution EM-density allow the positioning of individual subunits to produce a composite EMC structure (*Figure 5—figure supplement 5*; *Table 3*). The composite structural model accounts for the majority of EMC polypeptides with the exception of the TMD of EMC10 and flexible loops and termini of many subunits.

## The vestibule leads into a lipid-exposed intramembrane groove

A TMD substrate in the cytosolic vestibule of EMC needs to access the membrane interior for insertion. Furthermore, any route into the membrane should not only be accessible from the vestibule, but also be separated from the vestibule by a gap exposed to the cytosol. The gap is crucial for release of substrate from EMC; without it, the polypeptide segment between the substrate's TMD and cytosolic domain could not pass into the cytosol. This is analogous to the Sec61 complex, where a gap of ~10 Å separates the cytosolic end of the lateral gate and the ribosome surface surrounding the mouth of the polypeptide exit tunnel (*Voorhees et al., 2014*).

The cytosolic vestibule is continuous with a large horseshoe-shaped groove lined by the TMDs of EMC1, EMC3, EMC6, and EMC5 (*Figure 6A*; red star in *Figure 5A*). The lumenal side of this groove contains the crossbar contributed by EMC1. The groove stretches deep into the membrane and is

**Table 3.** EM refinement statistics.

| | |
|---|---|
| Resolution for refinement (Å) | 6.4 |
| CC$_{Mask}$ | 0.72 |
| RMS deviations | |
| Bond length (Å) | 0.005 |
| Bond angle (°) | 1.028 |
| No. of atoms | 9412 |
| Protein residues | 1906 |
| Ramachandran Outliers (%) | 0.49 |
| Mean B-factors (²) | 200.0 |
| MolProbity score | 2.77 |
| Clash score | 31.03 |
| PDB code | 6Z3W |

open laterally toward the lipid bilayer (*Figure 6B*,). Importantly, the cytosolic side of this groove at the N-terminus of EMC5 is separated from EMC2 by a clear gap of ~9 Å. This region of EMC2 is also predicted to be flexible and could move further away as needed during substrate insertion (*Figure 4—figure supplement 2*). Thus, a substrate TMD in the vestibule could potentially enter this groove and diffuse away from EMC with the cytosolic flanking polypeptide passing through the EMC2-EMC5 gap. Future mutagenesis experiments of groove-lining residues and site-specific cross-linking to substrates are needed to validate the proposed pathway of insertion.

## Discussion

TMDs are typically ~15–25 amino acid long with mostly hydrophobic side chains. Our analysis of EMC leads to a plausible mechanistic model for how it moves TMDs close to the N- or C-terminus of

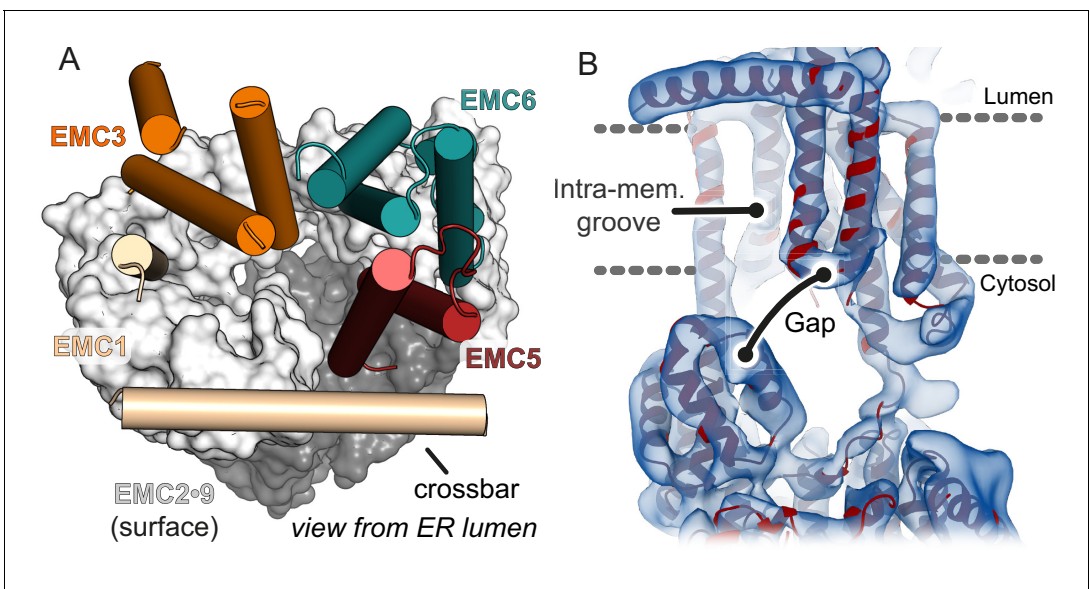

**Figure 6.** The cytosolic vestibule leads into an intramembrane groove. (**A**) The cytosolic vestibule in EMC2•EMC9 (grey surface) is contiguous with a horseshoe-shaped intramembrane groove lined by the TMDs of EMC1, EMC3, EMC5, and EMC6. The lumenal domain of EMC and the TMDs of EMC7 and EMC4 are not shown for clarity. (**B**) Fitting the composite EMC model (red) into cryo-EM density (blue) reveals a ~ 6–12 Å gap between the cytosolic and membrane domains of EMC. A substrate with a TMD in the intramembrane groove can be released from EMC with the cytosolic flanking polypeptide passing through this gap.

a protein from the aqueous cytosol to the membrane interior (*Figure 7A*). The partially hydrophobic character of a TMD favours binding to the cytosolic vestibule formed by the EMC2•EMC8 or EMC2•EMC9 complex. This vestibule is constitutively open to the cytosol, is shallow, and is only moderately hydrophobic. In these respects, EMC's vestibule differs from the targeting factors SRP and Get3, both of which have deeper, more hydrophobic grooves. These differences reflect their distinct roles: SRP and Get3 ferry TMDs through the cytosol to the membrane, while EMC engages TMDs that are already at the membrane. Thus, EMC2•EMC9 does not need to fully cover the TMD or bind to it stably; it instead serves as a portal of entry. Stable substrate binding to EMC2•EMC9 is not needed and might even be detrimental by impeding TMD release into the membrane.

EMC, like SRP and Get3, contains a conserved intramolecular 'placeholder' that sits in the substrate binding site and must be displaced by substrate. Hence, substrates that engage EMC2•EMC9 must not only be more favourable within the vestibule than in the aqueous environment, but also more favourable in the vestibule than the placeholder. These constraints may provide a degree of specificity for potential TMDs and discriminate against hydrophilic sequences. The conformation of substrates within the vestibule is not known, but might not be a helix in contrast to TMDs bound by SRP or Get3. This is because helix formation of a TMD occurs when shielding the polypeptide backbone is more favourable than exposing it to solvent. Because the vestibule is shallow, exposed to aqueous solvent, and has appreciable hydrophilic surfaces, the nascent TMD is likely to be unstructured when it first binds. This mode of binding might allow different regions of the vestibule to accommodate different types of TMDs. Furthermore, by exposing the backbone amides and carbonyls, even rather hydrophobic TMDs could be temporarily accommodated within a partially hydrophilic vestibule.

From the vestibule, the polypeptide has two points of egress: back toward the cytosol or into the intramembrane groove. This choice would be influenced by the polypeptide's properties. Hydrophilic polypeptides that transiently sampled the groove spuriously would diffuse back into the cytosol. Similarly, internal segments of polypeptides with large domains on either side would be sterically disfavoured from entering the groove. In this way, access into the groove serves as a filter that preferentially favors entry of terminal TMDs. Regardless of the precise register of the TMDs that form the intramembrane groove, its surface is more hydrophobic than the cytosolic vestibule. We therefore posit that polypeptide movement into the groove would be accompanied by helix formation. We expect substrate residence time in the groove would be minimal because it is open to the

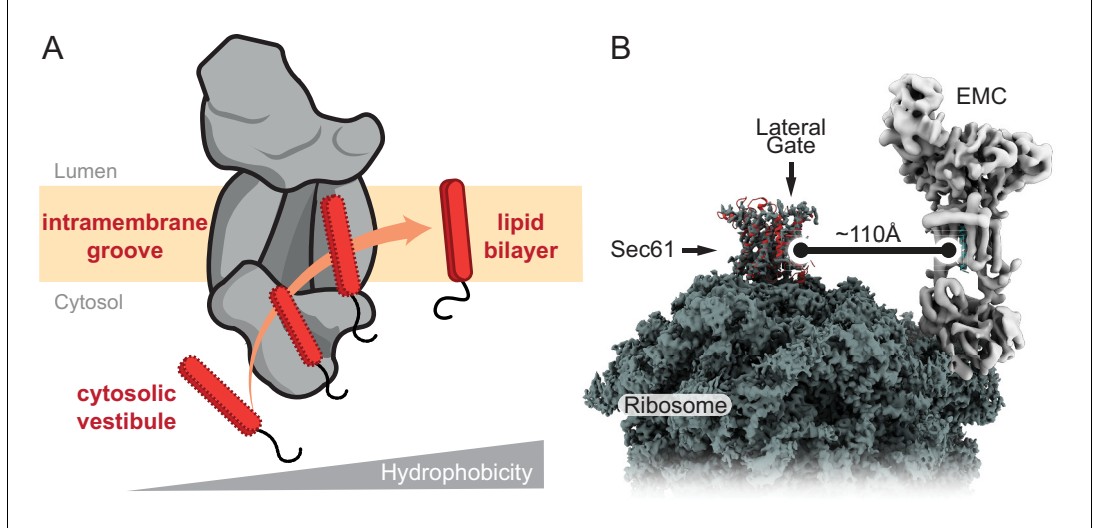

**Figure 7.** Model for EMC-mediated TMD insertion. (**A**) TMDs follow a gradient of increasing hydrophobicity from the cytosol through EMC and into the lipid bilayer. TMDs first enter the cytosolic vestibule formed by either the EMC2•EMC8 or EMC2•EMC9 complex. This vestibule leads into an intramembrane groove that is laterally open toward the lipid bilayer. (**B**) The ribosome-Sec61 complex is depicted with EMC at the same scale in the same membrane plane. The minimum distance from the Sec61 lateral gate to the intramembrane groove of EMC is ~110 Å. This implies that EMC can only mediate TMD insertion before ribosomes bind to Sec61 or if the polypeptide downstream of a TMD can span this distance. EMC cannot capture TMDs at the Sec61 lateral gate.

lipid bilayer. It is likely that TMD movement into the groove, helix formation, and partitioning into the membrane interior occur in rapid succession. Thus, the cytosol, vestibule, groove, and membrane interior are connected via a gradient of hydrophobicity that a TMD follows during its energy-independent insertion by EMC (*Figure 7A*).

The topology of the TMD would be influenced by steric constraints imposed by its flanking regions. Because the groove is not open toward the lumen, the side of the substrate containing only a short flexible segment can be translocated while the side with a bulky domain cannot. Translocation of the short terminus could be promoted despite the absence of a trans-bilayer pore by thermodynamically favorable release of a TMD into the lipid environment (*Engelman and Steitz, 1981*). This driving force can facilitate translocation of up to ~100 amino acid long polypeptide segments in the absence of a translocon (*Brambillasca et al., 2006*). In cases where both sides are short (e.g., very small single-TMD proteins), one orientation might be favoured over the other by flanking charges on the substrate being interpreted by charged residues on EMC.

It is noteworthy that EMC2 at the entry point into the intramembrane groove contains well-conserved solvent-exposed basic amino acids (see *Figure 2B*) such as Arg26 and Arg91. Furthermore, the ordered cytosolic tail of EMC1 at this key junction contains four conserved basic residues, at least some of which would face the vestibule. Finally, a conserved basic residue at position six in EMC5, and possibly its N-terminus, may also contribute net positive charge in this region. This enrichment of positive charge (and relatively fewer exposed negative charges) would disfavor entry into the groove of polypeptide segments that are enriched in basic amino acids. Such a mechanism would select against mitochondrial TA proteins, cleavable signal peptides, and $N_{cyt}$ signal anchors. In each case, these otherwise suitable EMC substrates (based on their terminal location and hydrophobicity) are enriched in basic amino acids in the short unstructured domain flanking the hydrophobic region (*Beltzer et al., 1991*; *Costello et al., 2017*; *von Heijne, 1985*; *Kalbfleisch et al., 2007*). Conversely, bona fide EMC substrates (ER-targeted TA proteins and $N_{exo}$ signal anchors) are typically dis-enriched in basic amino acids in the translocated tail. Thus, the mechanism of substrate selection by EMC might be part of the explanation for the 'positive-inside rule', a long-observed preference for positive flanking charges facing the cytosol (*Gafvelin et al., 1997*). This is an important area for future investigation.

The architecture and dimensions of EMC have major implications for when it acts during co-translational insertion of N-terminal TMDs and how it could function as a putative chaperone. EMC cannot feasibly approach closer than ~100 Å to a ribosome-bound Sec61 complex (*Figure 7B*). This means that EMC can act co-translationally on TMDs only before the ribosome docks on the Sec61 complex or only after at least ~100 Å of polypeptide downstream of the TMD has emerged from the ribosomal tunnel. Because ribosome-nascent chain complexes in which a TMD has just emerged from the ribosome can be inserted in an EMC-dependent manner, we have postulated that EMC acts after SRP-mediated targeting but before ribosomes bind to the Sec61 complex (*Chitwood and Hegde, 2019*). Due to SRP's position at the ribosome exit tunnel (*Halic et al., 2004*; *Schaffitzel et al., 2006*; *Voorhees and Hegde, 2015*), Sec61 cannot engage its binding site unless SRP releases the TMD and dissociates from the ribosome. Thus, there is a brief window when a TMD is available for EMC-mediated insertion before Sec61 binds to its position at the exit tunnel (*Jomaa et al., 2017*; *Kobayashi et al., 2018*). We posit that this time constraint helps minimize inappropriate insertion of N-terminal TMDs intended for the $N_{cyt}$ topology so they can be subsequently inserted by the Sec61 complex.

Although the intramembrane groove is well suited to serve as a chaperone for a single TMD, such a function seems unlikely to occur adjacent to the Sec61 translocon as proposed on the basis of ribosome profiling experiments (*Shurtleff et al., 2018*). Co-translational engagement of EMC by TMDs emerging from the ribosome-Sec61 complex could only occur once a TMD can diffuse at least 100 Å away. The very rigid structure of the EMC2•EMC9 complex and the absence of subcomplexes lacking these proteins makes it unlikely that the steric constraints shown in *Figure 7B* could be overcome. Furthermore, the intramembrane groove is sufficiently large to house one TMD, and possibly a second just outside the groove. This means that EMC is not likely to bind multiple TMDs simultaneously. Thus, a chaperone function would seem limited to post-translationally engaging an isolated TMD that has not yet assembled with its intra- or inter-molecular partners. Direct evidence for this function is currently lacking, but merits future study.

The role of the lumenal domain of EMC is currently unclear. The simplest possibility is that it serves a crucial structural role in stabilizing the seven integral membrane subunits while sealing the intramembrane groove toward the lumen. Maintaining the intramembrane groove configuration might be energetically unfavourable without structural caps on both sides. Other examples of proteins with a large lipid-exposed groove have most or all of their TMDs within a single polypeptide in contrast to EMC's multi-protein assembly (*Kumazaki et al., 2014*; *Ramasamy et al., 2013*; *Rollauer et al., 2012*; *Schoebel et al., 2017*). Thus, the lumenal domain, like EMC2, might nucleate EMC assembly, an early structural solution that was maintained across all eukaryotes.

Our proposed mechanism for EMC-mediated TMD insertion is conceptually similar to how the prokaryotic protein YidC is thought to function (*Dalbey et al., 2014*). YidC is a much simpler protein with a 5-TMD core whose arrangement forms an intramembrane 'hydrophobic slide' lined by TMD1, TMD2, and TMD5 (*Kumazaki et al., 2014*). The hydrophobic slide in YidC is appreciably smaller than the intramembrane groove in EMC and could not house a substrate TMD helix without conformational changes. Nevertheless, biochemical evidence supports the notion that substrate TMDs are near the hydrophobic slide during insertion (*Klenner et al., 2008*; *Yu et al., 2008*). Our structural model for the EMC3 subunit is consistent with an evolutionary relationship to YidC (*Anghel et al., 2017*). Notably however, EMC3 in our model is oriented such that the surface corresponding to YidC's hydrophobic slide is opposite to the surface lining EMC's intramembrane groove (see *Figure 5D*). This observation raises the possibility that the putative hydrophobic slide in EMC3 might be an alternative or additional route for substrate TMD insertion.

One way this could occur without risk of substrate entanglement in EMC is if the TMD approaches the hydrophobic slide from behind the EMC2•EMC9 vestibule. Upon insertion into the hydrophobic slide, the TMD would then release from EMC in the same direction from which it approached. Unlike in YidC however, access to lipid from the hydrophobic slide in EMC3 is partially occluded by EMC4 (*Figure 5D*). Thus, insertion via this alternative 'backside' route may require conformational changes to create more space between EMC3, EMC4, and EMC6 so substrates can enter the lipid bilayer.

Regardless of the exact route(s) of insertion through EMC, it appears that a simplified insertase originating from a YidC-like protein (*Borowska et al., 2015*) has been elaborated during evolution to form EMC. As Get1 appears to be a homolog of EMC3, the Get1/Get2 complex may be a simplified version of EMC that co-opted a different binding partner. High resolution structures of EMC's membrane region and the Get1/Get2 complex, together with structure-guided mutagenesis and substrate crosslinking experiments, will be important for fully elucidating and comparing their potentially shared mechanisms.

# Materials and methods

### Key resources table

| Reagent type (species) or resource | Designation | Source or reference | Identifiers | Additional information |
|---|---|---|---|---|
| Strain, strain background (*E. coli*) | BL21 NEB Express competent cells | New England Biolabs | Cat# C2523 | |
| Cell line (*Homo-sapiens*) | HEK293 TRex EMC5-FLAG | *Guna et al., 2018* | MBP04 | EMC5-FLAG (see below) integrated into FRT site of HEK293 TRex-Flp-in cell line. Adapted for growth in FreeStyle suspension media |
| Cell line (*Homo-sapiens*) | HEK293 TRex GFP-P2A-RFP-SQS | *Chitwood et al., 2018* | | |
| Cell line (*Homo-sapiens*) | HEK293 TRex OPRK-GFP-P2A-RFP | *Chitwood et al., 2018* | | |
| Cell line (*Homo-sapiens*) | HEK293T | ATCC | ATCC-CRL-3216 | |
| cell line (*Homo-sapiens*) | U2OS Flp-in TRex | *Volkmar et al., 2019* | | |

*Continued on next page*

*Continued*

| Reagent type (species) or resource | Designation | Source or reference | Identifiers | Additional information |
|---|---|---|---|---|
| Cell line (*Homo-sapiens*) | U2OS Flp-in TRex ΔEMC2 | *Volkmar et al., 2019* | | EMC2 disrupted using CRISPR-Cas9 |
| Antibody | ANTI-FLAG M2 Affinity Gel (mouse monoclonal) | Sigma | Cat# A4596 | |
| Antibody | Mouse monoclonal ANTI-FLAG M2-HRP conjugated | Sigma | Cat# A8592, RRID:AB_439702 | WB (1:10000) |
| Antibody | Rabbit polyclonal EMC1 | Thermo Fisher Scientific | Cat# A305-605A-M, RRID:AB_2782763 | WB (1:1000), IP (0.1 μg/sample) |
| Antibody | Rabbit polyclonal EMC4 | Thermo Fisher Scientific | Cat# A305-752A-M, RRID:AB_2782909 | IP (0.4 μg/sample) |
| Antibody | Rabbit polyclonal EMC4 (TMEM85) | Abcam | Cat# ab123719, RRID:AB_10951091 | WB (1:1000) |
| Antibody | Rabbit polyclonal EMC5 (MMGT1) | Abcam | Cat# ab174366, RRID:AB_2750837 | WB (1:1000) |
| Antibody | Rabbit polyclonal EMC6 (TMEM93) | Abcam | Cat# ab84902, RRID:AB_1925516 | WB (1:1000) |
| Antibody | Rabbit polyclonal EMC7 (C15orf24) | Proteintech | Cat# 27550–1-AP | WB (1:3000) |
| Antibody | Rabbit polyclonal EMC7 | Thermo Fisher Scientific | Cat# A305-678A-M, RRID:AB_2782836 | WB (1:1000) |
| Antibody | Rabbit polyclonal Calnexin N-terminus | Enzo Life Sciences | Cat# ADI-SPA-865, RRID:AB_10618434 | WB (1:5000) |
| Antibody | Rabbit polyclonal HA tag | This paper | Custom antibody raised against HA peptide conjugated to KLH. | WB (1:5000) |
| Recombinant DNA reagent | pET28a-6xHIS-SUMO-EMC2 | This paper | | Human EMC2, Residues 1–297 |
| Recombinant DNA reagent | pET21-EMC2-6xHIS | This paper | | Human EMC2, Residues 1–297 |
| Recombinant DNA reagent | pET28a-6xHIS-SUMO-EMC8 | This paper | | Human EMC8, Residues 1–210 |
| Recombinant DNA reagent | pET21-EMC9-6xHIS | This paper | | Human EMC9, Residues 1–208 |
| Recombinant DNA reagent | pET21-EMC2-6xHIS | This paper | | Human EMC2, Residues 11–274, for crystallization |
| Recombinant DNA reagent | pET21-EMC9-6xHIS | This paper | | Human EMC9, Residues 1–200 for crystallization |
| Recombinant DNA reagent | pET28a-6xHIS-ULP | This paper | | Yeast ULP1 protease (Uniprot: Q02724), Residues 403–621, for SUMO cleavage |
| Recombinant DNA reagent | SEC61-SQS (amber)–3 F4 | *Guna et al., 2018* | | T7-based PURExpress plasmid (New England Biolabs), SQS TMD residues 378–410, Amber mutation at F389 |
| Recombinant DNA reagent | SQS(amber)–3 F4 | This paper | | T7-based PURExpress plasmid (New England Biolabs), SQS TMD residues 378–410, Amber mutation at F389 |
| Recombinant DNA reagent | Clonetech-GFP-P2A-RFP-SQS | *Guna et al., 2018* | | |

*Continued on next page*

*Continued*

| Reagent type (species) or resource | Designation | Source or reference | Identifiers | Additional information |
|---|---|---|---|---|
| Recombinant DNA reagent | pcDNA5-EMC2-3xHA | This paper | | Human EMC2, Residues 1–297, Tet operator removed with Sac1 |
| Recombinant DNA reagent | pEVOL-pBpF | *Chin et al., 2002* | | |
| Recombinant DNA reagent | pcDNA5/FRT/TO-EMC3-Xamb-3xFLAG | This paper | | Human EMC3 with an amber codon at position X and a C-terminal 3xFLAG tag. Individual constructs where X = 23, 24, 25, 26, 71, 124, 125, 126, 127, 172, 173, 174, and 175 were produced. |
| Recombinant DNA reagent | pcDNA5/FRT/TO-EMC5-Xamb-3xFLAG | This paper | | Human EMC5 with an amber codon at position X and a C-terminal 3xFLAG tag. Individual constructs where X = 14, 15, 16, 17, 54, 55, 56, and 57 were produced. |
| Recombinant DNA reagent | pcDNA5/FRT/TO-3xFLAG-EMC6-Xamber | This paper | | Human EMC6 with an amber codon at position X and an N-terminal 3xFLAG tag. Individual constructs where X = 58, 59, 60, 61, 94, 95, 96, and 97 were produced. |
| Recombinant DNA reagent | pAS-Pyl-AF | This paper | | *Methanosarcina mazei* pyrrolysyl-tRNA synthetase with Y306A/Y384F mutations and its cognate tRNA carrying a U25C mutation |
| Recombinant DNA reagent | pcDNA5/FRT/TO-3xHA-EMC4 | This paper | | Human EMC4 with an N-terminal 3xHA tag |
| Recombinant DNA reagent | pcDNA5/FRT/TO-EMC4-3xHA | This paper | | Human EMC4 with a C-terminal 3xHA tag |
| Sequence-based reagent | siRNA #1 against EMC3 | Ambion | Custom synthesis | GGCACUAGAUGAUGUCGAAtt |
| Sequence-based reagent | siRNA #2 against EMC3 | Ambion | Custom synthesis | CCUACUAUGUUGACAGACAtt |
| Sequence-based reagent | siRNA #1 against EMC8 | Ambion | Custom synthesis | AGAUCAUAGCUACGUGAUUtt |
| Sequence-based reagent | siRNA #2 against EMC8 | Ambion | Custom synthesis | GCUGGUUAUUAUCAAGCUAtt |
| Sequence-based reagent | siRNA #1 against EMC9 | Ambion | Custom synthesis | GUACUUAUUAUGUUGGAUAtt |
| Sequence-based reagent | siRNA #2 against EMC9 | Ambion | Custom synthesis | AUGCAGCUGUGAACGAUCAtt |
| Peptide, recombinant protein | 3X FLAG Peptide | Sigma-Aldrich | Cat# F4799 | |
| Peptide, recombinant protein | Human SGTA | *Shao et al., 2017* | | |
| Peptide, recombinant protein | Human CaM | *Shao et al., 2017* | | |
| Peptide, recombinant protein | Human EMC2 | This paper | | Purified from NEB BL21 express cells |

*Continued on next page*

*Continued*

| Reagent type (species) or resource | Designation | Source or reference | Identifiers | Additional information |
|---|---|---|---|---|
| Peptide, recombinant protein | Human EMC8 | This paper | | Purified from NEB BL21 express cells |
| Peptide, recombinant protein | Human EMC9 | This paper | | Purified from NEB BL21 express cells |
| Peptide, recombinant protein | ULP1 Protease | This paper | | Purified from NEB BL21 express cells |
| Commercial assay or kit | PURE in vitro translation system | *Shao et al., 2017* | | |
| Chemical compound, drug | deoxy big CHAP (DBC) | Anatrace | Cat# 256455 | |
| Chemical compound, drug | Lauryl Maltose Neopentyl Glycol (LMNG) | Anatrace | Cat# NG310 5 GM | |
| Chemical compound, drug | Bpa | BACHEM | Cat# 4017646 | |
| Chemical compound, drug | AbK | Iris Biotech GmbH | Cat# HAA3110 | |
| Software, algorithm | CryoSPARC version 2.12.4. | *Punjani et al., 2017* | RRID:SCR_016501 | |
| Software, algorithm | UCSF ChimeraX Version 1.0 | *Goddard et al., 2018* | RRID:SCR_015872 | |
| Software, algorithm | trRosettta Structure Prediction Server | *Yang et al., 2020* | | |
| Software, algorithm | PHENIX Version 1.17 | *Adams et al., 2010*; *Afonine et al., 2018* | RRID:SCR_014224 | |
| Software, algorithm | *Coot* Version 0.9-pre | *Emsley et al., 2010* | RRID:SCR_014222 | |
| Software, algorithm | XDS Version 20190315 | *Kabsch, 2010* | RRID:SCR_015652 | |
| Software, algorithm | PyMol Version 2.1 | DeLano Scientific LLC | RRID:SCR_000305 | |
| Software, algorithm | DIALS Version 2.2 | *Winter et al., 2018* | | |
| Software, algorithm | BLEND CCP4i 7.0.076 | *Foadi et al., 2013* | | |
| Software, algorithm | Pointless CCP4i 7.0.076 | *Evans and Murshudov, 2013* | RRID:SCR_014218 | |
| Software, algorithm | Aimless CCP4i 7.0.076 | *Evans, 2011* | RRID:SCR_015747 | |
| Software, algorithm | SHELXD Version 2013/2 | *Sheldrick, 2008* | | |
| Software, algorithm | FlowJo Version 9.9.6 | FlowJo, LLC | RRID:SCR_008520 | |
| Software, algorithm | FlexEM | *Topf et al., 2008* | | |
| Other | 1.2/1.3 UltrAu Foil grids | Quantifoil | Direct Order Form System | |
| Other | Au 200 2/2 grids | Quantifoil | Direct Order Form System | |
| Other | HEK293 FreeStyle culture media | ThermoFisher Scientific | Cat# 12338001 | For HEK293 suspension adapted cell growth |

## Protein expression and purification

Bacterial over-expression plasmids for EMC2, EMC8, EMC9, SGTA, calmodulin, subsequent point mutations, and amber suppression mutations were produced by standard molecular biology techniques and listed in the Key Resources Table. Proteins were expressed in *E. coli* BL21 NEB express cells at 18°C for 18 hr following induction with 0.5 mM IPTG at an OD 600 of 0.6. For non-natural amino acid incorporation, amber codons were introduced at desired sites using site directed mutagenesis. Constructs were co-expressed with pEVOL-pBpF (*Chin et al., 2002*) in *E. coli* BL21 NEB express cells at 37°C. At OD 600 of 0.3, 0.2% L-arabinose was added to cultures and continued to grow to an OD 600 of 0.6 and cultures were cooled to 16°C. Once cultures were cooled, 0.15 mM IPTG and 1 mM benzoyl-phenylalanine (from a 1 M stock in 1 M NaOH) was added. Cells were then cultured at 16°C overnight. For both standard and amber suppression protein expression, cells were harvested via centrifugation, resuspended in ice cold Ni$^{2+}$-NTA buffer A (25 mM Tris pH 8.4, 500 mM NaCl, 20 mM Imidazole), lysed using sonication at 4°C, and subjected to centrifugation at 39,000 x g for 1 hr at 4°C. The supernatant containing hexahistidine-tagged proteins was passed over a column of Ni$^{2+}$-NTA matrix (Qiagen) at 0.5 mL matrix/1 L of culture. The matrix was washed with 20 column volumes of Ni$^{2+}$-NTA buffer A and eluted in a minimum of 3 column volumes using Ni$^{2+}$-NTA buffer A supplemented with imidazole to a final concentration 500 mM. Samples were buffer exchanged into 25 mM HEPES pH 7.5, 100–400 mM NaCl (depending on the protein) with either PD-10 or HiPrep 26/10 desalting columns (GE Healthcare). SGTA, calmodulin, and EMC amber suppression samples for in vitro assays were used directly and subsequently were supplemented with 10% glycerol and flash frozen in liquid nitrogen. EMC8 expression required an amino terminal SUMO solubilization tag, which was subsequently cleaved using ULP1 protease overnight at 4°C. Excess His-tagged SUMO and His-tagged ULP1 protease were removed via passage through a column of Ni$^{2+}$-NTA matrix. EMC2, EMC8, and EMC9 were concentrated and subjected to gel filtration using a GE 200 16/60 liquid chromatography column equilibrated in 25 mM HEPES pH 7.5 and 400 mM NaCl. Complex formation of EMC2•EMC8 or EMC2•EMC9 complexes was achieved by mixing a 1 to 1.2 molar ratio of EMC2 and either EMC8 or EMC9. Excess EMC8 or EMC9 was removed with gel filtration as indicated above.

## Size-Exclusion chromatography coupled to Multi-Angle light scattering

Protein samples at 1 mg/mL (15–40 mM) were injected onto an S200 10/300 Increase SEC column (GE, Marlborough, MA), equilibrated with 25 mM HEPES pH 7.5, 400 mM NaCl. The SEC column was coupled to a static 18-angle light scattering detector (DAWN HELEOS-II) and a refractive index detector (Optilab T-rEX) (Wyatt Technology, Goleta, CA). Data were collected every second at a flow rate of 0.5 mL/min. Data analysis was carried out using ASTRA VI, yielding the molar mass for each sample. The light scattering detectors were normalized and data quality were assessed by testing a BSA standard (Pierce).

## Microscale thermophoresis

Solvent exposed cysteine labeling of EMC2 was conducted with 100 µM EMC2 and 120 µM maleimide OG-488 (Molecular Probes, Eugene, OR) for 30 min on ice. Excess dye was removed from the labeling reaction with a NAP-5 column (GE, Marlborough, MA) pre-equilibrated with 25 mM HEPES pH 7.5, 500 mM NaCl. The labeling efficiency ranged from 55–65%. The concentration of EMC2•OG-488 was fixed at 50 nM and a dilution series of EMC8 or EMC9 ranging from 4 nM to 150 µM was used in the hydrophilic treated capillaries (NanoTemper Technologies). A Monolith NT.115 instrument (NanoTemper Technologies) was used with a 50% LED power and laser pulses at both 60% and 100% power. Data were analyzed using the NT Analysis software (NanoTemper Technologies).

## Crystallization of EMC2•EMC9 complex

EMC2•EMC9 complex comprising EMC2 residues 11–274 (of the 297 residue full length protein) and EMC9 residues 1–200 (of the 208 residue full length protein) were gel filtered into a buffer containing 25 mM Tris pH 8.5 and 400 mM NaCl. Protein at a concentration of 30 mg/mL was screened through sitting drop vapor diffusion using 200 nL protein mixed with 200 nL of reservoir solution and incubated at 20°C. Initial crystals were optimized via hanging drop vapor diffusion by mixing 800 nL

of protein (30 mg/mL) and 600 nL reservoir solution followed by incubation at 20°C. The crystal used for experimental phasing had a reservoir solution containing 28% PEG4000, 0.1 M Tris pH 8.3, and 0.2 M lithium sulfate. Crystals appeared and reached maximum dimensions without visual defects between 1–2 days. Crystals for experimental phasing were transferred to reservoir solution containing 20% glycerol for cryoprotection. Crystals used for native data set collection were grown in a reservoir containing 34% PEG4000, 0.1 M Tris pH 8.3, and 0.2 M lithium sulfate. Crystals for native collection were harvested within 24 hr directly from the drop by adding 4 μL of reservoir solution containing 20% ethylene glycol and incubating 1 min. Cryopreserved crystals were flash frozen in liquid nitrogen.

## Crystallographic data collection and analysis

Data were collected on EMC2•EMC9 crystals using I23 long-wavelength beamline at Diamond. Four native sulfur-SAD experiments with varied kappa goniometer orientation were conducted with a single crystal and each data set contained 3600 images collected at λ = 2.755 Å with 0.1° oscillations on a curved Pilatus 12M detector. Images were integrated and scaled with XDS and XSCALE (*Kabsch, 2010*). The substructure consisting of 27 sulfur sites was solved using SHELXD (*Sheldrick, 2008*) and phases were obtained with Autosol (*Adams et al., 2010*). The initial model was built manually and refined using *Coot* (*Emsley et al., 2010*) and PHENIX (*Adams et al., 2010*) respectively. The space group was P2₁2₁2₁ and a single EMC2•EMC9 molecule was observed in the asymmetric unit.

A native non-isomorphous P2₁2₁2₁ dataset was collected collected at I03 with a λ = 0.9763 Å, 3600 images with a 0.1° oscillation on an Eiger2 XE 16M detector. Data reduction was carried out with DIALS where four lattices offset by less than 5° were determined using multi-lattice processing (*Winter et al., 2018*), and each of the four lattices contained an equivalent number of reflections. The deconvolved datasets were analyzed in BLEND (*Foadi et al., 2013*) and merged/scaled using Pointless (*Evans and Murshudov, 2013*) and Aimless (*Evans, 2011*). The structure was determined by molecular replacement using the software package PHENIX (*Adams et al., 2010*). Iterative rounds of building and refinement were completed using *Coot* (*Emsley et al., 2010*) and PHENIX (*Afonine et al., 2018*), which resulted in the final model (PDB: 6Y4L). Data collection and refinement statistics are summarized in *Table 1*. Illustrations were made in Pymol versions 1.8.6 and 2.3.2 (Schrodinger, LLC). Surface chemical properties and conservation were displayed as previously described (*Ashkenazy et al., 2016*; *Hagemans et al., 2015*).

## Mammalian cell culture

All cell lines were cultured in DMEM containing 10% fetal calf serum and 2 mM L-glutamine. HEK293 T-REx Flp-In cells overexpressing EMC5-FLAG have been described (*Guna et al., 2018*). Wild type U2OS Flp-In TRex cells and ΔEMC2 U2OS Flp-In TRex cells have been characterized previously (*Volkmar et al., 2019*). HEK293T cells were originally from ATCC (ATCC CRL-3216). Cell line identities were verified by a combination of an integrated Frt site and doxycycline inducibility (distinctive to Flp-in T-Rex cells), by antibiotic resistance markers, and by immunoblotting for the product of knockout alleles. Further genetic analysis was not performed. Cells were checked approximately monthly for mycoplasma contamination using the MycoAlert Mycoplasma Detection Kit (Lonza) and found to be negative.

## Purification of native mammalian EMC

40 g pellets of suspension adapted HEK293 T-REx Flp-In cells overexpressing EMC5-FLAG were solubilized in 40 mL of 2x solubilization buffer [100 mM HEPES pH 7.4, 400 mM NaCl, 4 mM MgAc₂, 1.6% deoxy big CHAP (DBC; Merck – 256455)] for 1 hr with gentle shaking on ice. All subsequent steps apart from FLAG elution were conducted on ice or at 4°C. Solubilized cells were clarified for 20 min at 21,000 x g at 4°C and incubated for 1 hr with 1 mL bed volume of anti-FLAG M2 affinity gel (Sigma A4596) which had been pre-equilibrated in DBC wash buffer (0.3% DBC, 50 mM HEPES pH 7.4, 200 mM NaCl, 2 mM MgAc₂). FLAG resin was collected by centrifugation (5 min at 1500 x g) and washed twice with 4 mL DBC wash buffer. Resin was transferred to 10 mL gravity flow column and washed with 3 × 8 mL of 0.1% Lauryl Maltose Neopentyl Glycol (LMNG) wash buffer (0.1% LMNG, 50 mM HEPES pH 7.4, 200 mM NaCl, 2 mM MgAc₂) allowing 10 min between washes to

permit detergent exchange. Resin was then washed with 3 × 8 mL of 0.01% LMNG wash buffer (0.01% LMNG, 50 mM HEPES pH 7.4, 200 mM NaCl, 2 mM MgAc₂) allowing 10 min between washes for detergent exchange. Two FLAG elutions were conducted for 25 min each at room temperature by incubating the resin in 2 mL of FLAG elution buffer (0.01% LMNG, 50 mM HEPES pH 7.4, 200 mM NaCl, 2 mM MgAc₂, 0.25 mg/mL 3X FLAG peptide -Sigma F4799) with gentle end-over-end mixing. Combined FLAG elutions were diluted with 4 mL low-salt dilution buffer (0.01% LMNG, 50 mM HEPES pH 7.4, 2 mM magnesium acetate) and were bound to 150 µL bed volume of fast-flow SP sepharose, which had been pre-equilibrated with 2 mL ion exchange buffer A (0.01% LMNG, 50 mM HEPES pH 7.4, 50 mM sodium chloride, 2 mM magnesium acetate). SP sepharose was washed with 3 × 1 mL of ion exchange buffer A before 3 × 150 µL rounds of elution in ion exchange buffer B (0.01% LMNG, 50 mM HEPES pH 7.4, 400 mM sodium chloride, 2 mM magnesium acetate). The first elution was run over the resin twice and the resulting fractions were checked by A280. The peak fraction was then centrifuged for 30 min at 175,000 x g to remove insoluble aggregates.

## Cryo-EM sample preparation and data acquisition

Grid preparation consists of adding 3 µL of mammalian EMC complex at a concentration of 0.8–0.9 mg/mL to Quantifoil 1.2/1.3 UltrAu Foil grids or Quantifoil gold Au 200 2/2 grids. The Quantifoil 1.2/1.3 UltrAu Foil grids or Quantifoil gold Au 200 2/2 grids were first hydrophilized using a Fischione model 1070 plasma cleaner with a 9:1 ratio of Argon to Oxygen for 2 min and 30 s respectively. Grids were blotted for 9 s and plunged into liquid ethane using a Leica EM GP2 plunge freezer at 100% humidity. Particles were screened on a Technai F20 at 200 kV and data was collected on two FEI Titan Krios microscopes operating at 300 kV and equipped with Gatan K2 summit direct electron detectors, energy filters and volta phase plates. Data were collected at pixel sizes between 1.18 and 1.39 Å per pixel. Exposures were between 5 and 11 s long resulting total accumulated dose of 38 to 45 e⁻/Å² and were fractioned into 25 to 44 frames. All data were collected using FEI's EPU software at the MRC-LMB or eBIC.

## Cryo-EM data processing

Cryo-EM data processing was performed in the CryoSPARC processing suite (*Punjani et al., 2017*) using version 2.12.4. Initially five independent datasets were processed individually. Movies were motion corrected using multi-patch motion correction and CTF values estimated using multi-patch CTF estimation in CryoSPARC. Particles were picked using a blob picker before extraction with a box size of 224 pixels and 2D classification to produce templates for 2D-template based auto picking. Picked particles were subject to 2D and 3D reference-free classification and selected particles were refined using non-uniform refinement and local resolution estimation in CryoSPARC. Maps from datasets with differing nominal pixel sizes were then re-scaled to a range of pixel sizes and matched to a reference map in UCSF Chimera, yielding a cross correlation coefficient (*Tan et al., 2017*). Titration of these cross correlation values was used to identify an optimal nominal pixel size relative to the reference dataset (*Wilkinson et al., 2019*). The calculated values were used to re-scale the motion corrected micrographs from two datasets to match the nominal pixel size of 1.38 Å per pixel. The re-scaled datasets were then reprocessed as detailed above. Rescaled particles were merged with particles from three datasets originally collected at 1.38 Å per pixel, resulting in a master pool of 405,515 particles, which were subject to reference-free 3D classification. The final dataset of 167,294 particles was subject to non-uniform refinement followed by per-particle CTF refinement. CTF refined particles were subject to a final round of non-uniform refinement followed by local resolution estimation and filtering to produce a final map with a resolution of 6.4 Å. Refinement was limited to a maximum alignment resolution of 6.5 Å to prevent overfitting. During filtering a conservative *ad hoc* B-Factor of −150 was applied to the map to prevent over-sharpening. Full details of processing of individual and merged datasets are provided in *Table 2*.

## PURE in vitro translation reactions

A homemade PURE translation system (*Shimizu and Ueda, 2010*) was modified to accommodate amber-suppression as previously described (*Shao et al., 2017*). Changes that enable nearly quantitative amber-suppression are the omission of RF1, use of *E. coli* tRNA purified from a strain over-expressing the amber suppressor tRNA, addition of recombinant amber suppressor tRNA synthetase

(Bpa-RS) at 50 µg/mL, and addition of 0.1 mM of the photo-crosslinking non-natural amino acid benzoyl-phenylalanine (Bpa). Translation reactions of 25 µL were supplemented with 10 ng/µL DNA template, 24 µM CaM, 100 nM CaCl$_2$, and 2.5 µL $^{35}$S-methionine (from a stock of ~10 mCi/mL). Reactions were incubated at 37°C for 30 min, returned to ice, and diluted with 35 µL of ice-cold physiologic salt buffer (PSB: 50 mM HEPES pH 7.5, 100 mM KAc, 5 mM MgAc$_2$, 100 nM CaCl$_2$). Translated products were isolated via sucrose cushion by layering 50 µL of the diluted translation on top of 50 µL PSB supplemented with 20% sucrose. Centrifugation was conducted for 60 min at 55,000 RPM in a TLS-55 rotor with the slowest acceleration and deceleration possible. The top 80 µL of the sample were removed and snap frozen in liquid N$_2$ for storage prior to subsequent use in crosslinking reactions.

## Bpa crosslinking reactions

The translation products prepared as described above were diluted 1:4 in reaction buffer (25 mM HEPES pH 7.5, 500 mM NaCl, 100 nM CaCl$_2$) and mixed 1:1 with reaction buffer, SGTA, or EMC components at concentrations indicated in the figure legends. Samples were incubated at 37°C for 5 min, returned to ice and UV-irradiated using a UVP B-100 series lamp (UVP LLC) at a distance of 10 cm for a duration of 10 min.

## Protease protection assay

Plasmids encoding human EMC4 with an N- or C-terminal 3xHA tag in the pcDNA5/FRT/TO vector were transiently transfected into HEK293T cells using PEI 'MAX' (Polysciences). At 3 days after transfection, cells were collected, resuspended in ice-cold homogenizing buffer (20 mM HEPES pH 7.4, 250 mM sucrose, 2 mM MgCl$_2$, and EDTA-free protease inhibitor cocktail [Roche]), and homogenized on ice by 25 passages through a 26-guage needle fitted to a 1 mL syringe. Homogenates were centrifuged at 800 x g for 5 min at 4°C to pellet nuclei and debris, and the postnuclear supernatants were further centrifuged at 4000 x g for 10 min at 4°C to pellet heavy membranes. The supernatants were then subjected to ultracentrifugation at 100,000 x g for 30 min at 4°C, and the resulting microsome pellets were resuspended in ice-cold physiologic salt buffer-2 (50 mM HEPES pH 7.4, 100 mM KOAc, 2 mM MgAc$_2$) and placed on ice. Aliquots were left untreated or treated with 0.5 mg/mL of proteinase K in the presence or absence of 0.5% Triton X-100. After a 30 min incubation on ice, the digestion reactions were stopped by incubation with 5 mM PMSF on ice for 5 min and transferred to 10-volumes of boiling SDS lysis buffer (1% SDS and 0.1 M Tris-HCl pH 8.0). Samples were directly analyzed by SDS-PAGE and immunoblotting.

## Site-specific photo-crosslinking in mammalian cells

Position-specific incorporation of the photo-crosslinking amino acid AbK into proteins in mammalian cells was achieved by amber suppression using the *Methanosarcina mazei* pyrrolysyl-tRNA synthetase (PylRS) and tRNA$^{Pyl}_{CUA}$ pair (*Ai et al., 2011*). Plasmids used for the in vivo amber suppression were constructed as follows. The plasmid SE315 (*Oller-Salvia et al., 2018*), which encodes WT PylRS and the U25C mutant of tRNA$^{Pyl}_{CUA}$, was modified to express V5-tagged PylRS carrying Y306A and Y384F mutations [known as PylRS-AF mutant (*Yanagisawa et al., 2008*) instead of WT PylRS. The resultant plasmid was termed pAS-Pyl-AF and used throughout this study. The coding regions of human EMC subunits were inserted into a pcDNA5/FRT/TO vector containing an N-terminal or C-terminal 3xFLAG tag, and the codons for the selected positions were individually mutated to an amber codon in order to allow AbK incorporation. To express AbK-modified EMC subunits, HEK293T cells were co-transfected with pAS-Pyl-AF and an amber codon-containing EMC construct at a ratio of 4:1 using PEI 'MAX' (Polysciences) and were grown for 2 days in the presence of 0.5 mM AbK (Iris Biotech). After washing cells with PBS, ice-cold PBS was added, and photo-crosslinking was induced by placing cells on ice ~12.5 cm away from a UVP B-100 series lamp (UVP LLC) for 15 min. Cells were then harvested, frozen in liquid nitrogen, and stored at −80°C prior to analysis. For immunoprecipitation under non-denaturing conditions, cell pellets were lysed in native IP buffer (50 mM HEPES pH 7.4, 200 mM NaCl, 2 mM MgAc$_2$, 1% Triton X-100) supplemented with EDTA-free protease inhibitor cocktail (Roche). Cell lysates were cleared by centrifugation and incubated at 4°C for 1.5–2 hr with anti-FLAG M2 affinity gel (Sigma). After extensive washing with native IP buffer, proteins were eluted with SDS-PAGE sample buffer. Immunoprecipitation was also done under

denaturing conditions using antibodies against EMC subunits. For this, cell pellets were resuspended in SDS lysis buffer (1% SDS and 0.1 M Tris-HCl pH 8.0) and heat-denatured at 95°C for ~10 min with occasional vortex mixing. After cooling to room temperature, the denatured lysates were diluted 10-fold with IP buffer (50 mM HEPES pH 7.4, 100 mM NaCl, 1% Triton X-100) and incubated at 4°C for 1.5–2 hr with Protein A agarose beads plus the appropriate antibody, followed by extensive washing with IP buffer and elution with SDS-PAGE sample buffer.

## Flow cytometry analysis

Wild type U2OS Flp-In TRex cells and ΔEMC2 U2OS Flp-In TRex cells (*Volkmar et al., 2019*) were used in flow cytometry experiments and were maintained in DMEM supplemented with 10% fetal calf serum and 2 mM L-glutamine. Inverse transfections were conducted as follows: 3 µL GeneJuice (Merk Millipore) was added dropwise to 100 µL Opti-MEM (Gibco), vortexed and incubated at room temperature for 5 min. SQS-reporter DNA (250 ng) and either EMC2 DNA or empty vector (750 ng) were added to the Opti-MEM/GeneJuice mixture, solution was pipetted up and down to mix, and incubated at room temperature for 20 min. Cells were seeded into a six well dish at 40% confluency concurrent with the addition of 100 µL of the transfection/DNA mixture. Cells were grown for 3 days and then detached with trypsin/EDTA in 250 µL volume, neutralized with 500 µL complete media, and pelleted by centrifugation at 5000 x g in a table top centrifuge at 4°C. Cells were washed by resuspension in 500 µL ice cold PBS and re-pelleted as above. Pelleted cells were resuspended in a final volume of 600 µL ice cold PBS and filtered through a 70 µm cell strainer for analysis. Cells were analyzed using a BD LSR II flow cytometer (BD Biosciences, Franklin Lakes, NJ). The flow cytometry experiment and data analysis using FlowJo software packages were conducted as previously described (*Itakura et al., 2016*). Knockdowns of EMC3, EMC8 and EMC9 with siRNAs were for 3 days in HEK 293 Trex cells with stably integrated inducible reporters (GFP-P2A-RFP-SQS or OPRK-GFP-P2A-RFP) using Lipofectamine RNAiMAX reagent (Thermo Fisher). siRNA sequences are available in the Key Resources Table). Expression of the reporters was induced with 1 µg/ml doxycycline for ~20 hr before analysis by flow cytometry performed as described above.

## Statistics and reproducibility

This study did not employ any statistical analysis or comparisons, no grouping, and no power calculations. Biochemical and cell biological experiments include internal controls that were performed as part of the same experiment and are shown within the respective figures. Each result or conclusion from any given figure panel represents one example of at least two biologically independent experiments, both of which contained the same or similar internal controls. No attempts at replication with valid controls failed. Where data are compiled from multiple measurements (*Figure 2—figure supplement 2*), the number of replicates are indicated in the figure legend with error bars to indicate standard deviation.

## Acknowledgements

We thank P Chitwood for helpful discussions, MM Babu for co-evolution advice, RL Williams and D Bellini for advice on crystallography analysis, J Chin and I Gregor for amber suppression reagents, A Guna for help with protein preparations at an early stage of this project, N Volkmar and J Christianson for discussions and the ΔEMC2 cells, G Cannone for EM support, J Grimmett and T Darling for scientific computing, the EM, crystallization, mass spectrometry, and biophysics core facilities at the LMB, and the staff on beamlines I03 and I23 at the Diamond Light Source for assistance with crystal screening and data collection. We acknowledge access and support of the Cryo-EM facilities at the UK national electron bio-imaging centre (eBIC), proposal EM17434-54, funded by the Wellcome Trust, MRC and BBSRC. We acknowledge Diamond Light Source for beamtime under proposal MX21426. Funding was provided by the Medical Research Council of the UK (MC_UP_A022_1007 to RSH and MC_UP_1201/10 to EAM) and the US National Institutes of Health (R01 GM078186 to EAM and R01 GM130051 to RJK). JPO was funded by an EMBO long term fellowship (ALTF 18–2018), BPP was supported by Boehringer Ingelheim Fonds, and YY was funded by fellowships from the Naito Foundation and the Osamu Hayaishi Memorial Scholarship for Study Abroad. DM acknowledges the support of the Swiss National Science foundation (SNF) under grant P2ELP3_18910.

# Additional information

## Competing interests

Ramanujan S Hegde: Reviewing editor, *eLife*. Elizabeth A Miller: Reviewing editor, *eLife*. The other authors declare that no competing interests exist.

## Funding

| Funder | Grant reference number | Author |
|---|---|---|
| Medical Research Council | MC_UP_A022_1007 | Ramanujan S Hegde |
| Medical Research Council | MC_UP_1201/10 | Elizabeth A Miller |
| National Institutes of Health | R01 GM078186 | Elizabeth A Miller |
| National Institutes of Health | R01 GM130051 | Robert J Keenan |
| European Molecular Biology Organization | ALTF 18-2018 | John P O'Donnell |
| Boehringer Ingelheim Fonds | | Ben Photon Phillips |
| Naito Foundation | | Yuichi Yagita |
| Japanese Biochemical Society | | Yuichi Yagita |
| Swiss National Science Foundation | P2ELP3_18910 | Duccio Malinverni |

The funders had no role in study design, data collection and interpretation, or the decision to submit the work for publication.

## Author contributions

John P O'Donnell, Conceptualization, Funding acquisition, Investigation, Methodology, Writing - original draft, Writing - review and editing, Performed the biochemical and biophysical analysis of EMC cytosolic subunits, solved the EMC2•EMC9 crystal structure, analyzed EMC mutants in cells, and contributed to model generation, building, and interpretation; Ben P Phillips, Conceptualization, Investigation, Methodology, Writing - review and editing, Initiated and performed the cryo-EM analysis (with help from JPO at later stages) and contributed to the building and interpretation of models; Yuichi Yagita, Investigation, Methodology, Writing - review and editing, Set up the site-specific photo-crosslinking assay in cells and analysed EMC4 topology; Szymon Juszkiewicz, Investigation, Writing - review and editing, Performed flow cytometry analysis of EMC subunit knockdowns; Armin Wagner, Duccio Malinverni, Investigation, Methodology, Writing - review and editing; Robert J Keenan, Formal analysis, Methodology, Writing - review and editing; Elizabeth A Miller, Conceptualization, Supervision, Funding acquisition, Project administration, Writing - review and editing; Ramanujan S Hegde, Conceptualization, Supervision, Funding acquisition, Writing - original draft, Project administration, Writing - review and editing

## Author ORCIDs

John P O'Donnell ⓘ https://orcid.org/0000-0002-0571-3603
Ben P Phillips ⓘ https://orcid.org/0000-0001-9117-4695
Szymon Juszkiewicz ⓘ http://orcid.org/0000-0002-3361-7264
Armin Wagner ⓘ http://orcid.org/0000-0001-8995-7324
Robert J Keenan ⓘ http://orcid.org/0000-0003-1466-0889
Elizabeth A Miller ⓘ http://orcid.org/0000-0002-1033-8369
Ramanujan S Hegde ⓘ https://orcid.org/0000-0001-8338-852X

## Decision letter and Author response

Decision letter https://doi.org/10.7554/eLife.57887.sa1
Author response https://doi.org/10.7554/eLife.57887.sa2

## Additional files

### Supplementary files
• Transparent reporting form

### Data availability

Structural coordinates have been deposited in PDB under accession codes 6Y4L and 6Z3W. Cryo-EM data had been deposited to EMDB under accession code EMD-11058. All other data in this study are provided within the manuscript.

The following datasets were generated:

| Author(s) | Year | Dataset title | Dataset URL | Database and Identifier |
|---|---|---|---|---|
| O'Donnell JP, Wagner A, Phillips BP, Hegde RS | 2020 | Crystal structure of the human EMC2•EMC9 complex | https://www.rcsb.org/structure/6Y4L | RCSB Protein Data Bank, 6Y4L |
| O'Donnell JP, Phillips BP, Hegde RS | 2020 | Cryo-EM structure of human EMC | https://www.rcsb.org/structure/6Z3W | RCSB Protein Data Bank, 6Z3W |
| Hegde RS, Phillips BP, O'Donnell JP, Miller EA | 2020 | Human ER Membrane protein Complex (EMC) | https://www.ebi.ac.uk/pdbe/emdb/11058 | Electron Microscopy Data Bank, 11058 |

The following previously published datasets were used:

| Author(s) | Year | Dataset title | Dataset URL | Database and Identifier |
|---|---|---|---|---|
| Voorhees RM, Hegde RS | 2016 | The structure of the mammalian Sec61 channel opened by a signal sequence | https://www.rcsb.org/structure/3JC2 | RCSB Protein Data Bank, 3JC2 |
| Voorhees RM, Fernandez IS, Scheres SHW, Hegde RS | 2014 | Structure of the idle mammalian ribosome-Sec61 complex | https://www.rcsb.org/structure/3J7Q | RCSB Protein Data Bank, 3J7Q |

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
