## [Decision Letter]

**Acceptance summary:**

The manuscript describes a very detailed structural analysis of the EMC complex that is involved in incorporating N- or C-terminal transmembrane helices into the ER membrane. It describes a combination of x-ray crystallography and cryo-EM as well as cross linking techniques to establish a model of the entire complex despite the relatively low resolution which requires the use of extensive modelling. Overall, the entire procedure is carefully performed and provides an excellent model that explains the structure and function of this important complex.

**Decision letter after peer review:**

Thank you for submitting your article "The architecture of EMC reveals a path for membrane protein insertion" for consideration by *eLife*. Your article has been reviewed by three peer reviewers, including Volker Dötsch as the Reviewing Editor and Reviewer #1, and the evaluation has been overseen by John Kuriyan as the Senior Editor.

The reviewers have discussed the reviews with one another and the Reviewing Editor has drafted this decision to help you prepare a revised submission.

Summary:

In their manuscript Dr Ramanujan S. Hegde and co-workers address the structure and molecular mechanism of EMC, an ER membrane protein complex which comprises ten subunits and facilitates the insertion of some tail anchored (TA)- and certain multipass-membrane proteins. In the latter case only the most aminoterminal transmembrane domain appears to be head first-/Nexo-inserted by EMC, additional domains of the same membrane proteins were suggested to be integrated into the membrane by the Sec61 complex. Therefore, EMC was proposed to mediate membrane insertion of either amino- or carboxy-terminal transmembrane domains which cannot be handled by the Sec61 complex.

The authors employed state of the art methods for the purification of native EMC as well as individual subunits or mutant variants thereof, 3D reconstruction after cryo-electron microscopy, X-ray crystallography, site specific crosslinking, and functional plus additional biochemical assays for the characterization of the purified proteins.

Revisions:

1) The RMSD value should be given in A, not A squared.

2) Among the reviewers the evidence for the inner membrane cavity part of the EMC being a slide for TMH integration was discussed. For your information the original suggestion for experimental evidence is included here:

"To be convincing, the second part needs to be supported by functional data, such as crosslinking of at least the TA model protein to the intramembrane groove (e.g. at low temperature) plus rational design of mutant variants of groove subunits and test of their membrane insertion capacity for the TA model protein in intact cells. Furthermore, even in the presence of such results it would be difficult to unequivocally exclude the chaperone model for EMC function. In my opinion, the latter would require either crosslinking or 3D reconstruction after cryo-EM of nascent multipass-membrane proteins to EMC and Sec61 complex, and the use of EMC mutant variants in studies on the biogenesis of multipass membrane proteins in intact cells, respectively."

However, in the discussion it was agreed that no further experiments are needed, but that the authors should make more careful statements (i.e. use the words suggest, hypothesize, etc.) and that a sentence that "substrate cross linking will be used for validation of the intramembrane groove in future work" should be added.

3) Subsection “Functional analysis of the vestibule in the EMC2•EMC9 complex”. It is stated that EMC8 or EMC9 alone suppress aggregation of the model TA substrate less than EMC2 alone does. Whilst true, the difference is not huge for EMC8 (roughly two fold difference) so it seems unsafe to conclude that only EMC2 has specific interactions with the substrate. In fact we later find that EMC9/EMC8 forms part of the substrate binding cavity. It is stated in the Figure 1—figure supplement 2 legend that the crosslinks between the TA substrate and EMC2 alone are physiologically relevant but those between the substrate and EMC8 and EMC9 are due to co-aggregation with the substrate. How do the authors know this?

4) Figure 3. The HLY substitution that perturbs model substrate integration includes the substitution of an amino acid Y191 that has previously been used as a negative control for the BPA crosslinking experiments because it faces away from the binding cavity. It's therefore very difficult to infer that substitution of this residue is affecting substrate binding by changing the polarity of the cavity (rather than changing the protein fold).

5) Figure 1—figure supplement 2. The CaM-substrate adduct should be labelled to assist the reader.

6) Figure 5B. The rotation between the two views appears to be the opposite direction to that indicated in the figure. I appreciate that this figure is included primarily to show the quality of the model fit to the EM density, but I did struggle to assign the subunits by their colours under the EM density overlay. For example I could not identify EMC1. It might be worth labelling the subunits nearest the viewer on the figure.

7) Figure 5C. It would be helpful to know where the crosslinking substituted amino acids are on the structure in Figure 5D. Similarly, it would be useful to have the identify (aa number) of all the crosslinking amino acids shown in Figure 5D marked on the figure.

8) Figure 5D. Where is the raw data for this figure for those substitutions not shown in Figure 5C? I do think it is misleading to draw a crosslinking line from the substituted amino acid to an exact position on the crosslinked partner subunit because the position of the acceptor amino acid has not been confirmed experimentally. Minimally the legend should say that the crosslink is drawn to the closest point in the model (rather than saying `verified crosslinks'), but perhaps on the figure these could be shown as arrows not quite reaching the partner subunit.

9) Figure 1—figure supplement 3. The authors should indicate in the legend that both of the proteins analysed are EMC substrates.

10) Figure 4. The placeholder density referred to in the subsection “Position of the EMC2•EMC9 subcomplex within native EMC” should be indicated on the figure if it is present. If it is not this should be made clear when referring to it in the text. The text should also refer forward to where this density is discussed in detail (or even better all this discussion should be moved to where the structure of the density is analysed in Figure 5).

11) Figure 5E. Again I appreciate that this panel is to show the fit of the model to the EM density but I would like to be able to see the model more clearly. I recommend that the authors include another panel/figure showing the model of the lumenal domain in different views.

12) Figure 5—figure supplement 3. The TM part of EMC3 should be marked because it is not obvious which helices the modelled EMC3 cytoplasmic density should be associated with. Is there support for the fold of the EMC3 three-helix bundle from a co-evolution model? The raw crosslinking data should be shown.

13) Are there significant parts of the EMC polypeptides that are not included in the model that has been fitted to the EM density?

---

## [Author Response]

Revisions:1) The RMSD value should be given in A, not A squared.

Fixed.

2) Among the reviewers the evidence for the inner membrane cavity part of the EMC being a slide for TMH integration was discussed. For your information the original suggestion for experimental evidence is included here:"To be convincing, the second part needs to be supported by functional data, such as crosslinking of at least the TA model protein to the intramembrane groove (e.g. at low temperature) plus rational design of mutant variants of groove subunits and test of their membrane insertion capacity for the TA model protein in intact cells. Furthermore, even in the presence of such results it would be difficult to unequivocally exclude the chaperone model for EMC function. In my opinion, the latter would require either crosslinking or 3D reconstruction after cryo-EM of nascent multipass-membrane proteins to EMC and Sec61 complex, and the use of EMC mutant variants in studies on the biogenesis of multipass membrane proteins in intact cells, respectively."However, in the discussion it was agreed that no further experiments are needed, but that the authors should make more careful statements (i.e. use the words suggest, hypothesize, etc.) and that a sentence that "substrate cross linking will be used for validation of the intramembrane groove in future work" should be added.

We agree, and this important caveat has been added (subsection “The vestibule leads into a lipid-exposed intramembrane groove”), as well as more cautious language throughout. We have also added an additional paragraph about alternative models of insertion at the end of the revised Discussion.

3) Subsection “Functional analysis of the vestibule in the EMC2•EMC9 complex”. It is stated that EMC8 or EMC9 alone suppress aggregation of the model TA substrate less than EMC2 alone does. Whilst true, the difference is not huge for EMC8 (roughly two fold difference) so it seems unsafe to conclude that only EMC2 has specific interactions with the substrate. In fact we later find that EMC9/EMC8 forms part of the substrate binding cavity. It is stated in the Figure 1—figure supplement 2 legend that the crosslinks between the TA substrate and EMC2 alone are physiologically relevant but those between the substrate and EMC8 and EMC9 are due to co-aggregation with the substrate. How do the authors know this?

We deleted the line suggesting EMC2 is the major contributor to substrate interaction. The primary basis on which we judged whether the crosslinks are functionally relevant (versus non-specific, potentially due to co-aggregation) was whether the substrate-substrate interaction was simultaneously diminished. This is now clarified in the legend.

4) Figure 3. The HLY substitution that perturbs model substrate integration includes the substitution of an amino acid Y191 that has previously been used as a negative control for the BPA crosslinking experiments because it faces away from the binding cavity. It's therefore very difficult to infer that substitution of this residue is affecting substrate binding by changing the polarity of the cavity (rather than changing the protein fold).

We have added a caveat (subsection “Functional analysis of the vestibule in the EMC2•EMC9 complex”) noting that in addition to hydrophobicity, the mutations might alter the local conformation to account for the phenotype. Of course, the overall fold is not likely to be perturbed given this mutant protein effectively assembles with the rest of EMC.

5) Figure 1—figure supplement 2. The CaM-substrate adduct should be labelled to assist the reader.

This label has been added.

6) Figure 5B. The rotation between the two views appears to be the opposite direction to that indicated in the figure. I appreciate that this figure is included primarily to show the quality of the model fit to the EM density, but I did struggle to assign the subunits by their colours under the EM density overlay. For example I could not identify EMC1. It might be worth labelling the subunits nearest the viewer on the figure.

We apologise this figure was not clearer. The rotation is correct and can be seen by the position of the crossbar moving from the plane of the page to a perpendicular position on the left side. To further clarify this, we have labelled the front-facing subunits as recommended.

7) Figure 5C. It would be helpful to know where the crosslinking substituted amino acids are on the structure in Figure 5D. Similarly, it would be useful to have the identify (aa number) of all the crosslinking amino acids shown in Figure 5D marked on the figure.

We tried extensively to incorporate amino acid labels but it was not possible without making the figure extremely complicated or having the labels be very small. Instead, we have prepared a supplementary figure (Figure 5—figure supplement 3) that shows the primary data for the crosslinking, and in panel F lists the amino acid numbers of the key crosslinking sites depicted on the structural model.

8) Figure 5D. Where is the raw data for this figure for those substitutions not shown in Figure 5C? I do think it is misleading to draw a crosslinking line from the substituted amino acid to an exact position on the crosslinked partner subunit because the position of the acceptor amino acid has not been confirmed experimentally. Minimally the legend should say that the crosslink is drawn to the closest point in the model (rather than saying `verified crosslinks'), but perhaps on the figure these could be shown as arrows not quite reaching the partner subunit.

We tried arrows but it was hard to find a clear view where all the arrowheads would be simultaneously visible. We therefore edited the legend as suggested. The primary data documenting the crosslinking are now shown in Figure 5—figure supplement 3.

9) Figure 1—figure supplement 3. The authors should indicate in the legend that both of the proteins analysed are EMC substrates.

Good point. This has been added.

10) Figure 4. The placeholder density referred to in the subsection “Position of the EMC2•EMC9 subcomplex within native EMC” should be indicated on the figure if it is present. If it is not this should be made clear when referring to it in the text. The text should also refer forward to where this density is discussed in detail (or even better all this discussion should be moved to where the structure of the density is analysed in Figure 5).

This is now indicated in the figure and the text edited accordingly. In line with the overall suggestion to be more cautious, we refer to this as a ‘potential placeholder’ to indicate that our assignment to EMC6 is provisional.

11) Figure 5E. Again I appreciate that this panel is to show the fit of the model to the EM density but I would like to be able to see the model more clearly. I recommend that the authors include another panel/figure showing the model of the lumenal domain in different views.

We agree that it is difficult to see the model through the density. Therefore, a supplementary figure has been added that shows views of the complete EMC model without density (Figure 5—figure supplement 5).

12) Figure 5—figure supplement 3. The TM part of EMC3 should be marked because it is not obvious which helices the modelled EMC3 cytoplasmic density should be associated with. Is there support for the fold of the EMC3 three-helix bundle from a co-evolution model? The raw crosslinking data should be shown.

The three-helix bundle is predicted by co-evolution and this is now noted in the text (subsection “Architecture of the membrane-embedded and lumenal regions of EMC”). The crosslinking data is now provided as part of Figure 5—figure supplement 3. Dashed lines are now drawn between the cytosolic and membrane regions of EMC3.

13) Are there significant parts of the EMC polypeptides that are not included in the model that has been fitted to the EM density?

The main parts that are not accounted in the model are the TM domain of EMC10, and the presumably flexible loops and termini of most subunits. This is now noted at the end of the subsection “Architecture of the membrane-embedded and lumenal regions of EMC”.